# PCAInit: Training-Free Initialization for Image-Based Neural Representations

## Abstract

Implicit neural representations (INRs) have been widely used to model data as continuous functions parameterized by multi-layer perceptrons (MLPs). However, the relationship between the weight space of INRs and the underlying data space remains underexplored. In this paper, using SIREN as a baseline architecture, we study this connection through the lens of video frame reconstruction, which serves as a controlled setting where principal component analysis (PCA) reveals a striking alignment between image space and weight space. Building on this observation, we introduce *PCAInit*, a novel training-free initialization strategy. We compare PCAInit with pretrained-based approaches that also offer higher reconstruction quality but come at the cost of additional training time: a meta-learned initialization and our two additional proposed methods. We show that PCAInit achieves the best overall reconstruction quality without extra training time. For example, on a representative DAVIS 2017 video (bear, 480p), PCAInit improves PSNR by up to +37.1% over SIREN and +26.7% over meta-learned initialization. Furthermore, we show that PCAInit generalizes beyond video frames, achieving the best PSNR on collections of images as well. Moreover, we demonstrate that PCAInit achieves high PSNR in additional evaluation tasks and exhibits strong universality through cross-video initialization experiments. Our results reveal a promising research direction on the interplay between image space and weight space in INRs, opening new avenues for future research on efficient INRs with improved reconstruction quality and broader applicability.

## 1 Introduction

Implicit neural representations (INRs), or coordinate-based neural representations, have emerged as a powerful paradigm for diverse modalities (Tewari et al., 2022; Essakine et al., 2025), including images, audio, and 3D shapes (Sitzmann et al., 2020b), as well as neural rendering methods such as NeRFs (Mildenhall et al., 2020). Rather than storing data explicitly, INRs map coordinates to discrete signals (e.g., images, audio, or 3D shapes) through multi-layer perceptrons (MLPs), enabling compact storage, continuous interpolation, and high-fidelity reconstruction. Beyond signal representation, recent works have also used INRs themselves as data for downstream tasks such as classification (Dupont et al., 2022), point cloud segmentation and retrieval (Luigi et al., 2023), and 3D shape generation (Erkoç et al., 2023), introducing the broader research direction of deep weight space (Navon et al., 2023)[1].

Despite their wide applications, training the underlying MLPs remains challenging. SIREN (Sitzmann et al., 2020b) is widely adopted thanks to its ability to capture fine details across multiple modalities, but it requires a carefully designed initialization to ensure stable training and convergence. Several improvements have been proposed to enhance reconstruction quality (Essakine et al., 2025), including novel activation functions (Ramasinghe & Lucey, 2022; Saragadam et al., 2023; Saratchandran et al., 2024; Liu et al., 2024) and positional encodings (Tancik et al., 2020). However, the fundamental cost of training a new network instance from scratch for each individual signal is still unsolved.

---

[1]In the following, we use the term 'weight' to denote the model parameters, for consistency with the notion of weight space.

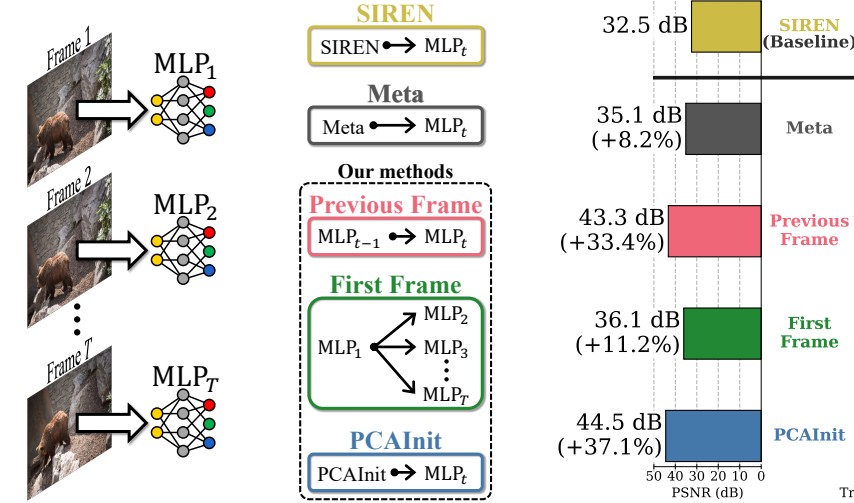

(a) Fitting a single video frame with an MLP

(b) Initialization methods

(c) Trade-off between PSNR and training time of all methods on the *bear* video, using SIREN as the baseline with gains reported relative to it

Figure 1: **Overview**: (a) our goal is to fit a single image with one Multi-Layer Perceptron (MLP); (b) we propose new initialization methods for MLPs; (c) mean PSNR across all frames and the total training time on the DAVIS 2017 *bear* video at 480p resolution. Compared to the SIREN baseline, the Meta initialization achieves better PSNR but requires substantially longer total training time when accounting for both the meta-training and test-time adaptation phases. Our Previous Frame method surpasses the Meta initialization in PSNR, though at the cost of higher training time due to its sequential nature. The First Frame method, in turn, reduces training time relative to Previous Frame but yields lower image quality. Finally, our PCAInit method achieves the highest PSNR while requiring the same training time as the SIREN baseline, clearly outperforming all other initialization methods in both quality and efficiency.

A promising direction to reduce training time is to develop initialization strategies that acceler-ate convergence across diverse instances. Meta-learned initialization methods explore this idea by training networks on large collections of signals (e.g., 3D models (Sitzmann et al., 2020a) or im-ages (Tancik et al., 2021)) to obtain transferable weights that provide strong initializations for unseen instances. But these methods remain limited to small-scale settings, such as low-resolution images, due to memory and computational requirements.

In order to overcome the aforementioned challenges, we study the weight initialization problem in the context of video frame reconstruction, where our investigation primarily focuses on two di-rections: (a) pretraining-based approaches (e.g., meta-learned initialization) and (b) hand-designed initialization schemes (e.g., SIREN). For (a), we propose two initialization methods that reuse pre-trained models and improve over the meta-learned baseline in reconstruction quality. Nevertheless, these methods still incur additional training cost, therefore, we investigate (b), where we seek a training-free initialization that does not require any additional training time.

Our investigation in the direction of (b) reveals an intriguing phenomenon where we discover a relation between image space and weight space in INRs. Based on this discovery, we introduce PCAInit, a novel training-free initialization method that directly initializes network weights, similar to SIREN, but achieves substantially higher reconstruction quality without any additional training.

To sum up, our main contributions are summarized as follows:

- We propose two initialization strategies that leverage pretrained models for INR training on video frames, improving reconstruction quality compared to the SIREN baseline.
- We introduce PCA initialization, a novel training-free initialization method derived from an analysis of the relation between image space and weight space. Our method achieves the best image quality without additional training time.

- We show that PCA initialization generalizes beyond video data by applying it to collections of images, opening new research directions on the relation between image space and weight space of INRs.

## 2 RELATED WORK

**Implicit Neural Representations.** Early works such as Occupancy Networks (Mescheder et al., 2019) and DeepSDF (Park et al., 2019) focused mainly on 3D shape representation. Later, SIREN (Sitzmann et al., 2020b) was introduced as a versatile method that works well across multiple modalities, enabled by a tailored initialization scheme that ensures stable training and convergence. Around the same time, Fourier Features (Tancik et al., 2020) were proposed to address the spectral bias of MLPs, enabling them to capture high-frequency variations in signals. Subsequent research explored improved activation functions to extend SIREN, including Gauss (Ramasinghe & Lucey, 2022), WIRE (Saragadam et al., 2023), Sinc (Saratchandran et al., 2024), and Finer (Liu et al., 2024). For broader overviews, we refer readers to the excellent surveys of Tewari et al. (2022), Xie et al. (2022), and Essakine et al. (2025).

For videos, NeRV (Chen et al., 2021), its successor NeRV-Enc (Chen et al., 2024), and NIR-VANA (Maiya et al., 2023) achieve compression performance comparable to conventional video codecs. We would like to emphasize that, although we introduce our setting through video frame reconstruction, our primary goal in this paper is to study the weight initialization problem for INRs. We use video frame reconstruction primarily as a testbed for the analysis in Section 3.3, and an in-depth comparison to INR-based video representation methods is therefore beyond the scope of this work. Furthermore, these methods employ hybrid architectures that combine INRs with convolutional or transformer networks, making the underlying representation difficult to analyze in isolation. For this reason, we focus on high-fidelity image reconstruction and leverage the architectural simplicity and strong representational power of SIREN (Sitzmann et al., 2020b).

**Meta-learning Initialization.** As briefly discussed in Section 1, meta-learning algorithms such as MAML (Finn et al., 2017) and Reptile (Nichol et al., 2018) have been applied to INR training. In this setting, the meta-training phase learns general-purpose weights from a large collection of signals, and the test-time adaptation phase corresponds to training an INR on a new signal instance with only a few gradient updates. The two most well-known approaches are MetaSDF (Sitzmann et al., 2020a) for signed distance fields and Meta-Initialization (Tancik et al., 2021) for multiple modalities. In the context of video frame reconstruction, while these methods accelerate convergence, the total training time, including both the meta-training phase and the short test-time adaptation phase, is often longer than training SIREN from scratch.

Motivated by the idea of reusing pretrained models, our first two methods adopt this strategy to initialize INR training and thereby improve reconstruction quality. Unlike meta-learning approaches, we do not have the dedicated meta-training phase; however, the pretraining step itself still introduces additional computational overhead. To remove this overhead entirely, our third method analyzes the intrinsic structure of the INR weight space and its relation to the signal being represented. This analysis connects directly to prior work on representational alignment, which we discuss next.

**Weight Space and Representational Alignment.** A growing body of work seeks to understand neural network representations by analyzing the geometry and relationships within their output features, latent spaces, or the weights themselves (Sucholutsky et al., 2023; Huh et al., 2024). Prior studies have largely focused on representational similarity in latent spaces. For instance, some analyze angle preservation in latent features (Moschella et al., 2023) or provide comprehensive surveys of similarity measures (Klabunde et al., 2025). Others investigate cross-modal alignment, such as the relationship between the outputs of 3D and text encoders (Hadgi et al., 2025). In parallel, several studies examine the weight space directly. These efforts include analyzing flattened weights for downstream performance prediction (Dupont et al., 2022; Luigi et al., 2023) and exploring permutation symmetries, either across layers (Zhou et al., 2023; 2024; Navon et al., 2023; 2024) or through graph-based formulations (Kofinas et al., 2024; Lim et al., 2024).

Diverging from these directions, we introduce a novel perspective: analyzing the alignment between the INR weight space and the signal domain. Specifically, our method projects both the weight space and the image space into a comparable low-dimensional basis using principal component analysis

(PCA). To our knowledge, this is the first work to directly study how the structure of INR weights aligns with the structure of the signals they represent. We elaborate on this approach in the following section.

# 3 METHODS

We first introduce the background of video frame reconstruction using INRs in Section 3.1. Next, we describe two initialization strategies in Section 3.2 and discuss their limitations. We then analyze the relationship between image and weight spaces in Section 3.3. This analysis motivates our third initialization method, PCAInit, presented in Section 3.4. Finally, in Section 3.5, we extend PCAInit from video frames to collections of independent images.

## 3.1 IMAGE AND VIDEO FITTING WITH SIREN

We begin with the task of image fitting using INRs. Given an image $\mathbf{I} \in \mathbb{R}^{H \times W \times 3}$, we represent it as a continuous function $f$ that maps a 2D coordinate to an RGB value. This function is approximated by an MLP $f_\theta$, parameterized by weights $\theta$, where $f_\theta : \mathbb{R}^2 \to \mathbb{R}^3$ takes a continuous pixel coordinate $\mathbf{x}$ and outputs its corresponding color value $f(\mathbf{x})$. In this work, we use SIREN (Sitzmann et al., 2020b), which employs sine activations as periodic functions. Image coordinates are normalized to the range $[-1, 1]^2$, and RGB values are rescaled from $[0, 255]$ to $[-1, 1]$.

As illustrated in Figure 1a, for a video $\{\mathbf{I}_t\}_{t=1}^T$ with $T$ frames, we approximate each frame function $f_t$ independently using an MLP $f_{\theta_t}$. The training objective is

$$\mathcal{L}(\theta_t) = \sum_{\mathbf{x} \in \Omega} \|f_{\theta_t}(\mathbf{x}) - f_t(\mathbf{x})\|_2^2,$$

where $\Omega$ denotes the set of normalized pixel coordinates on the $H \times W$ grid:

$$\mathbf{x}_{ij} = \left( \frac{2i}{H-1} - 1, \frac{2j}{W-1} - 1 \right), \quad i \in \{0, \dots, H-1\}, \ j \in \{0, \dots, W-1\}.$$

Since video frames can be extracted independently, training an MLP for each frame can be parallelized. For example, using a single NVIDIA RTX A6000 GPU, five MLPs corresponding to five frames can be simultaneously trained, reducing the time to train substantially.

## 3.2 PRETRAINING INITIALIZATION

In our setup, we describe weight initialization as $\theta^{(A)} \leftarrow \theta^{(B)}$, where $\theta^{(B)}$ denotes the source weights used for initialization, and $\theta^{(A)}$ is the resulting set of MLP weights. In this scenario, $\theta_t$ is used to denote an untrained MLP for frame $t$ and $\theta_t^*$ denotes its optimized counterpart. With that in mind, we first investigate two initialization strategies illustrated in Figure 1b and describe them below.

**Previous Frame.** For frame $t$, we initialize $\theta_t$ with the optimized weights of the MLP trained using the previous frame, $\theta_t \leftarrow \theta_{t-1}^*$, and the first frame is initialized separately using SIREN initialization: $\theta_1 \leftarrow \theta^{\text{SIREN}}$. This strategy improves convergence and reconstruction quality, but it forces strictly sequential training, making it computationally expensive.

**First Frame.** Alternatively, we initialize all later frames with the optimized weights of the first frame, $\theta_t \leftarrow \theta_1^*$ for all $t \geq 2$. This enables parallel training but results in lower performance compared to the Previous Frame strategy.

Based on the results in Figure 1c, the trade-off between reconstruction quality and parallelism remains substantial, highlighting the need for a better initialization strategy. Motivated by the observation that the Previous Frame method achieves a clear PSNR improvement over the baseline SIREN, we hypothesize that there exists a relationship between video frames and their corresponding optimized weights. This motivates a deeper analysis of the alignment between image space and weight space, which we discuss next.

We note that our Previous Frame initialization is conceptually related to NIRVANA (Maiya et al., 2023). However, NIRVANA fits SIREN representations in a feature space, and the resulting features are then fed to a NeRV-like block to produce RGB values, whereas we fit SIREN directly in the image space, which is more straightforward.

### 3.3 MOTIVATION: IMAGE–WEIGHT SPACE ALIGNMENT

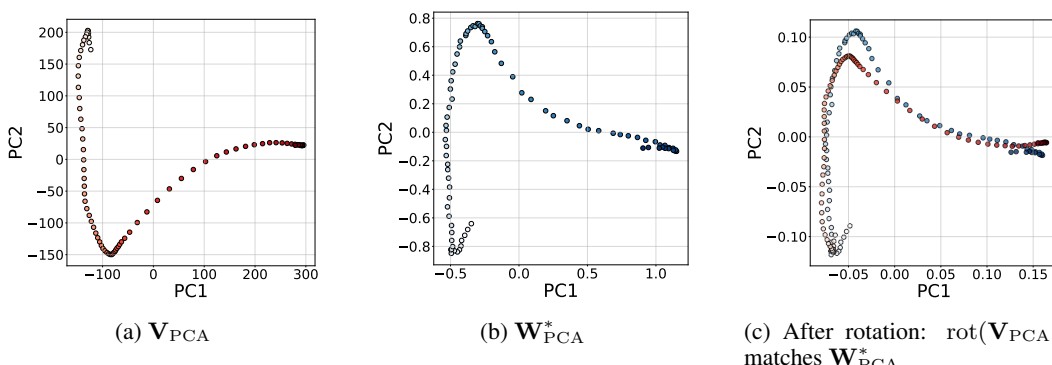

(a) $\mathbf{V}_{\mathrm{PCA}}$        (b) $\mathbf{W}^*_{\mathrm{PCA}}$        (c) After rotation: $\mathrm{rot}(\mathbf{V}_{\mathrm{PCA}})$ matches $\mathbf{W}^*_{\mathrm{PCA}}$

Figure 2: **Image–Weight space alignment.** After training MLPs with SIREN initialization on the DAVIS 2017 *bear* video at 480p resolution: (a) Principal component trajectory of flattened video frames $\mathbf{V}_{\mathrm{PCA}}$. Darker shades of red correspond to later frames ($t$), indicating the temporal order. (b) Principal component trajectory of flattened optimized weights $\mathbf{W}^*_{\mathrm{PCA}}$. Darker shades of blue correspond to later frames ($t$), indicating the temporal order. (c) After Procrustes rotation, $\mathrm{rot}(\mathbf{V}_{\mathrm{PCA}})$ aligns closely with $\mathbf{W}^*_{\mathrm{PCA}}$, with a Procrustes disparity of $0.021$, confirming structural similarity.

After training MLPs with SIREN initialization on the DAVIS 2017 *bear* video at 480p resolution, we analyze the relationship between video frames and their optimized weights. To this end, we flatten both representations and compare their trajectories under PCA projection. Specifically, we perform the following process.

**Video frames.** Each frame $\mathbf{I}_t$ is flattened into $\mathrm{flat}(\mathbf{I}_t) \in \mathbb{R}^{(H \cdot W \cdot 3)}$, and the results are stacked:

$$\mathbf{V} = \{\mathrm{flat}(\mathbf{I}_1), \mathrm{flat}(\mathbf{I}_2), \dots, \mathrm{flat}(\mathbf{I}_T)\} \in \mathbb{R}^{T \times (H \cdot W \cdot 3)}.$$

Applying PCA to $\mathbf{V}$ yields

$$\mathbf{V}_{\mathrm{PCA}} \in \mathbb{R}^{T \times d},$$

where $d = 2$ principal components are retained for visualization and analysis in our study.

**MLP weights.** Similarly, each trained MLP $\theta_t^*$ is flattened into $\mathrm{flat}(\theta_t^*) \in \mathbb{R}^{n_\theta}$, and the results are stacked:

$$\mathbf{W}^* = \{\mathrm{flat}(\theta_1^*), \mathrm{flat}(\theta_2^*), \dots, \mathrm{flat}(\theta_T^*)\} \in \mathbb{R}^{T \times n_\theta},$$

where $n_\theta$ is the total number of trainable parameters in the MLP. Applying PCA to $\mathbf{W}^*$ gives

$$\mathbf{W}^*_{\mathrm{PCA}} \in \mathbb{R}^{T \times d}.$$

As illustrated in Figure 2, visualizations of $\mathbf{V}_{\mathrm{PCA}}$ and $\mathbf{W}^*_{\mathrm{PCA}}$ show that the two trajectories appear strikingly similar, even across ten different random seeds. To confirm this quantitatively, we employ Procrustes analysis (Schönemann, 1966), which finds the optimal orthogonal transformation (rotation and reflection) that aligns one set of points with another while minimizing squared error.[2] The computed Procrustes disparity between $\mathbf{V}_{\mathrm{PCA}}$ and $\mathbf{W}^*_{\mathrm{PCA}}$ confirms strong alignment, with consistently low scores (e.g., $0.021$ on the DAVIS 2017 *bear* video at 480p resolution). Moreover, applying this procedure to other DAVIS 2017 videos consistently reveals the same image–weight alignment phenomenon (in Appendix A.2). These findings suggest that trajectories in image space and weight space are structurally aligned.

---

[2] We use the `orthogonal_procrustes` implementation from SciPy (Virtanen et al., 2020).

One might hypothesize that $\text{flat}(\theta_t^*)$ could be predicted directly from $\mathbf{V}_{\text{PCA}}$ via a rotation. However, this would require access to optimized weights $\theta_t^*$, which are not available at initialization time. This motivates the development of a new initialization method that leverages the observed alignment, which we present in the next subsection.

## 3.4 PCAINIT: PCA-BASED INITIALIZATION VIA IMAGE–WEIGHT SPACE ALIGNMENT

To exploit the observed alignment without needing optimized weights, we construct a pseudo weight trajectory. First, we initialize one MLP with SIREN initialization, denoted $\theta^{\text{SIREN}}$, and duplicate it $T$ times:

$$\mathbf{W}^{\text{SIREN}} = \{\text{flat}(\theta^{\text{SIREN}}), \dots, \text{flat}(\theta^{\text{SIREN}})\} \in \mathbb{R}^{T \times n_\theta}.$$

Since all rows are identical, this trajectory is degenerate (i.e., PCA is uninformative). We therefore perturb each copy with random noise by adding an element-wise offset

$$\epsilon \sim \mathcal{U}[-r, r], \quad r = 10^{-3}, \quad \epsilon \in \mathbb{R}^{T \times n_\theta},$$

to obtain

$$\tilde{\mathbf{W}}^{\text{SIREN}} = \mathbf{W}^{\text{SIREN}} + \epsilon \in \mathbb{R}^{T \times n_\theta}.$$

This perturbation serves only to break the degeneracy and does not inject any meaningful signal from the data. We selected $r = 10^{-3}$ based on a validation experiment on the DAVIS 2017 *bear* video, and further analyze the effect of different noise ranges $r$ in Appendix A.3.2.

Running PCA on $\tilde{\mathbf{W}}^{\text{SIREN}}$ yields

$$\tilde{\mathbf{W}}^{\text{SIREN}}_{\text{PCA}} = \tilde{\mathbf{W}}^{\text{SIREN}}\mathbf{P} \in \mathbb{R}^{T \times d},$$

where $\mathbf{P} \in \mathbb{R}^{n_\theta \times d}$ is the PCA basis matrix. We then compute the Procrustes transformation $\mathbf{R} \in \mathbb{R}^{d \times d}$ aligning $\mathbf{V}_{\text{PCA}} \in \mathbb{R}^{T \times d}$ with $\tilde{\mathbf{W}}^{\text{SIREN}}_{\text{PCA}}$, and obtain

$$\text{rot}(\mathbf{V}_{\text{PCA}}) = \mathbf{V}_{\text{PCA}}\mathbf{R} \in \mathbb{R}^{T \times d},$$

where $\text{rot}(\mathbf{V}_{\text{PCA}})$ denotes the transformed image trajectory aligned to weight space.

Finally, we use the PCA basis $\mathbf{P}$ to unproject $\text{rot}(\mathbf{V}_{\text{PCA}})$ back into the original $n_\theta$-dimensional weight space

$$\mathbf{W}^{\mathbf{V}} = \text{rot}(\mathbf{V}_{\text{PCA}})(\mathbf{P})^\top \in \mathbb{R}^{T \times n_\theta}.$$

Equivalently, this yields initialization weights

$$\mathbf{W}^{\mathbf{V}} = \{\text{flat}(\theta_t^{\mathbf{V}})\}_{t=1}^T \in \mathbb{R}^{T \times n_\theta},$$

where $\theta_t^{\mathbf{V}}$ denotes the initialization for frame $t$ derived from video $\mathbf{V}$. Each $\theta_t^{\mathbf{V}}$ is then used to initialize $\theta_t$, thus enabling parallel training while preserving quality. We call this initialization strategy PCAInit.

**Summary.** The overall procedure can be summarized as

$$\mathbf{V} \in \mathbb{R}^{T \times (H \cdot W \cdot 3)} \;\mapsto\; \mathbf{V}_{\text{PCA}} \in \mathbb{R}^{T \times d} \;\mapsto\; \text{rot}(\mathbf{V}_{\text{PCA}}) \in \mathbb{R}^{T \times d} \;\mapsto\; \mathbf{W}^{\mathbf{V}} \in \mathbb{R}^{T \times n_\theta}.$$

In short, PCAInit rotates the low-dimensional video trajectory into weight space and then lifts it back to full dimensionality to produce initialization weights. The temporary noise-injected weights $\tilde{\mathbf{W}}^{\text{SIREN}}$ are used only to provide the Procrustes transformation $\mathbf{R}$ and the valid PCA basis $\mathbf{P}$ for unprojection, which overcomes the limitation of not having access to the optimized weights $\theta_t^*$ at initialization time.

## 3.5 BEYOND VIDEO FRAMES: APPLICATION TO IMAGE COLLECTIONS

Instead of consecutive video frames, PCAInit can also be applied to a set $\{\mathbf{I}_i\}_{i=1}^N$ of unrelated images (e.g., the first frame from $N$ different videos). In this case, the image stack plays the same role as a video sequence: PCA is computed across the flattened images to capture dominant variations in the collection, and these trajectories are then aligned and unprojected into weight space to yield initialization weights for each image:

$$\mathbf{W}^{\{\mathbf{I}_i\}} = \{\text{flat}(\theta_i^{\{\mathbf{I}_i\}})\}_{i=1}^N \in \mathbb{R}^{N \times n_\theta}.$$

Here, $\theta_i^{\{\mathbf{I}_i\}}$ denotes the initialization for image $\mathbf{I}_i$ obtained from the PCAInit procedure. Note that, in the video case, we used the notation $\mathbf{W}^{\mathbf{V}}$ to denote weights derived from a sequence of frames, while for image collections we use $\mathbf{W}^{\{\mathbf{I}_i\}}$ to avoid ambiguity. We also conduct an additional experiment for the case $N{=}1$ (a single image) in Appendix A.6.

## 4 EXPERIMENTS

**Dataset.** We conduct our experiments on the DAVIS 2017 dataset (Pont-Tuset et al., 2017), which contains 90 diverse natural videos at $480p$ resolution. For our study, we carefully select 15 representative videos: *bear*, *bike-packing*, *boxing-fisheye*, *breakdance-flare*, *crossing*, *dog-agility*, *dog-gooses*, *drift-chicane*, *gold-fish*, *kid-football*, *lab-coat*, *parkour*, *planes-water*, *snowboard*, and *tennis*. The selection is guided by two key factors: (1) the type of camera movement and (2) the most dominant color in the scene. This choice allows us to systematically examine whether motion characteristics and visual appearance influence the behavior of our initialization methods.

In particular, several videos emphasize distinct characteristics: *gold-fish* and *planes-water* are dominated by blue scenes, *dog-gooses* contains extensive green regions, *snowboard* is characterized by white backgrounds with fast movements, *boxing-fisheye* exhibits strong lens distortion, and *lab-coat* features repetitive zoom-in/zoom-out patterns. These diverse conditions provide a challenging yet controlled benchmark for assessing the generality of our proposed initialization strategies. A complete overview of the selected videos is provided in Appendix A.1.

**Baselines.** We compare our proposed initialization method against several widely used approaches. We first use SIREN (Sitzmann et al., 2020b) as the primary baseline. We also include positional encoding approaches, which apply an additional mapping layer after the input: FF (Tancik et al., 2020), which employs a Gaussian mapping layer, and NeRF (Mildenhall et al., 2020), which adds a positional encoding layer. Finally, we evaluate Meta (Tancik et al., 2021), a meta-learned initialization trained on large collections of images.

**Implementation details.** For each video frame, we train an MLP following the baseline setup. Unless otherwise specified, all MLPs consist of 3 hidden layers with 512 hidden units, resulting in a model size of approximately 3.1 MB. All networks use sine activations, except FF and NeRF, which use ReLU activations. We adopt the AdamW optimizer from PyTorch with default settings, a learning rate of $1 \times 10^{-4}$, and train for 1000 epochs. In each epoch, we use a single batch containing all pixel coordinates on the $H \times W$ grid, so that one optimization step corresponds to one epoch.

For Meta, we follow the MAML setting with 150,000 iterations, inner learning rate = $1 \times 10^{-2}$, outer learning rate = $1 \times 10^{-5}$, and inner steps = 2. Due to GPU memory constraints, we set the batch size to 1 for this method. The meta-training phase is performed on all frames of a single video.

All models are trained on a single NVIDIA RTX A6000 GPU (48 GB) without mixed precision. We ensure fair comparison by keeping the architecture and optimization settings consistent across all initialization methods.

**Metrics.** To evaluate reconstruction quality, we use PSNR (Peak Signal-to-Noise Ratio) and SSIM (Structural Similarity Index Measure). We also report the total training time, measured as the elapsed wall-clock time from start to completion, to analyze efficiency. Together, these metrics provide complementary perspectives on reconstruction quality and computational cost.

### 4.1 VIDEO-FRAME EVALUATION

**Single-video evaluation.** We use the *bear* video (Figure 1) as a representative case. The PSNR convergence is shown in Figure 3, qualitative reconstructions in Figure 4, and quantitative results (mean and standard deviation of PSNR and SSIM) in Table 1.

In Table 1, FF and NeRF achieve the lowest results, while SIREN is moderate. Meta improves over SIREN but remains below our methods, indicating that meta-learning is less practical for video frames compared to First Frame, which provides a strong initialization with minimal training cost.

Previous Frame and PCAInit obtain the highest PSNR and SSIM. While Previous Frame is strong, its sequential dependence prevents parallelization and leads to long training time (Figure 1c). A

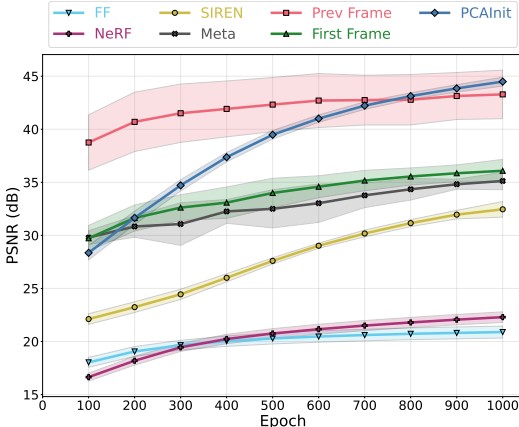

Figure 3: Mean and standard deviation of PSNR across frames per epoch on the *bear* video.

Table 1: Mean and standard deviation of PSNR and SSIM across frames on the *bear* video. Best results in **bold**, second-best underlined.

| Method | PSNR (dB) ↑ | SSIM ↑ |
|---|---|---|
| FF | $20.89 \pm 0.56$ | $0.354 \pm 0.020$ |
| NeRF | $22.30 \pm 0.50$ | $0.434 \pm 0.014$ |
| SIREN | $32.46 \pm 0.73$ | $0.923 \pm 0.024$ |
| Meta | $35.13 \pm 0.82$ | $0.952 \pm 0.014$ |
| Prev Frame | $\underline{43.29 \pm 2.29}$ | $\underline{0.988 \pm 0.007}$ |
| First Frame | $36.09 \pm 1.07$ | $0.959 \pm 0.014$ |
| PCAInit | $\mathbf{44.49 \pm 0.42}$ | $\mathbf{0.992 \pm 0.001}$ |

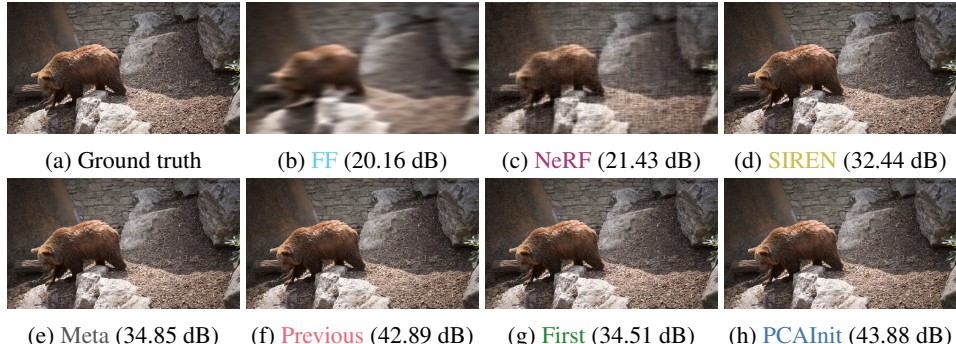

(a) Ground truth  (b) FF (20.16 dB)  (c) NeRF (21.43 dB)  (d) SIREN (32.44 dB)

(e) Meta (34.85 dB)  (f) Previous (42.89 dB)  (g) First (34.51 dB)  (h) PCAInit (43.88 dB)

Figure 4: Reconstruction of the ground-truth last frame ($82^{nd}$) of the *bear* video using different methods. For readability, we shorten Previous Frame and First Frame to Previous and First, respectively. Our PCAInit achieves the highest fidelity with 43.88 dB PSNR, clearly outperforming all baselines.

chunked parallelization strategy could reduce this cost but would likely degrade PSNR, so we do not adopt it. Thus, PCAInit is the most practical method, balancing accuracy and efficiency.

As shown in Figure 3, PCAInit initially lags behind Meta, First Frame, and Previous Frame, but quickly surpasses the first two by 300 epochs and outperforms all methods by 1000 epochs. PCAInit also yields the lowest standard deviation, reflecting stable initialization, whereas universal approaches show higher variance and Previous Frame exhibits large std due to gradual frame-to-frame improvements (e.g., the last-frame PSNR is much higher than the first-frame PSNR). Figure 4 further confirms this trend, showing that PCAInit produces the most faithful reconstruction of the 82nd frame with 43.88 dB PSNR, clearly outperforming all baselines.

**Multi-video evaluation.** We perform frame reconstruction for each video, compute the average results across all 15 videos, and plot the PSNR histogram of all frames across all videos in Figure 5. We exclude FF and NeRF due to their low performance in the single-video experiment. We also exclude Meta because of its high training cost. Thus, we focus on the most competitive and efficient methods for multi-video evaluation.

Our PCAInit achieves the highest stability across all videos, combining strong mean PSNR (45.35 dB) with the lowest variance, while Previous Frame attains slightly higher mean PSNR (45.37 dB) but suffers from instability due to its sequential dependence. The narrow histogram of PCAInit further highlights its robustness across diverse conditions, including varying motion and appearance.

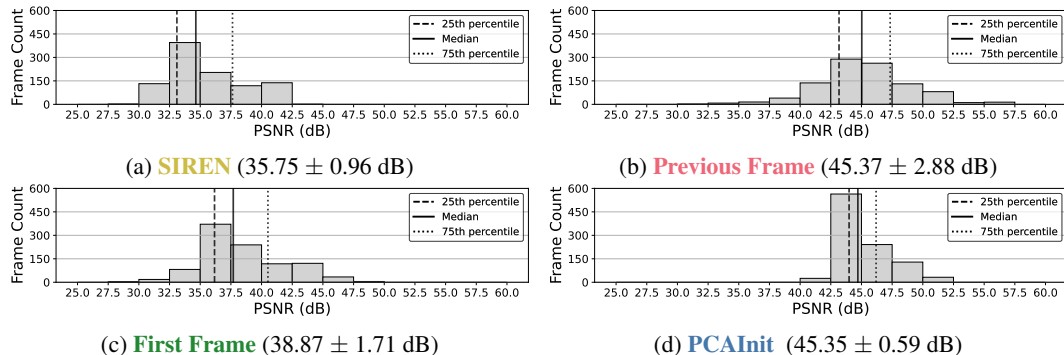

Figure 5: PSNR distribution of all frames across the 15 DAVIS videos for different initialization methods. Sub-captions report mean ± standard deviation (dB). Our PCAInit achieves a competitive mean PSNR with the lowest variance, while Previous Frame yields the highest mean PSNR but with much larger variance due to its sequential dependence. The narrow histogram of PCAInit confirms its robustness across diverse motion and appearance conditions.

Table 2: Mean and standard deviation of PSNR and SSIM results for the image-collection experiment. The highest scores are shown in **bold**. Our PCAInit achieves the best reconstruction quality with large margins over all baselines, highlighting its robustness even when images are not sequentially related.

| Method | PSNR (dB) ↑ | SSIM ↑ | Total Time (hours) ↓ |
|---|---|---|---|
| SIREN | $35.67 \pm 3.05$ | $0.937 \pm 0.037$ | 3.51 |
| Meta | $37.44 \pm 3.36$ | $0.954 \pm 0.023$ | 59.56 |
| Previous Frame | $39.47 \pm 3.33$ | $0.968 \pm 0.012$ | 17.10 |
| First Frame | $38.18 \pm 3.50$ | $0.963 \pm 0.014$ | 4.64 |
| PCAInit | $\mathbf{45.19 \pm 2.00}$ | $\mathbf{0.991 \pm 0.003}$ | 3.87 |

## 4.2 IMAGE-COLLECTION EVALUATION

Recall Section 3.5, we collect the first frame from each of the 15 selected videos to create an image stack $\in \mathbb{R}^{15 \times 480 \times 854 \times 3}$ and conduct experiments in the same manner as in Section 4.1. Note that we resize the *bike-packing* frame from $(480, 910)$ to $(480, 854)$ to match the resolution of the other videos.

From Table 2, we observe that SIREN performs the worst, while PCAInit substantially outperforms all methods. Specifically, PCAInit improves over Meta by +7.75 dB PSNR, and even surpasses Previous Frame by +5.72 dB PSNR. This demonstrates that Previous Frame only works well when there is strong correlation among images (e.g., consecutive video frames). In contrast, PCAInit remains effective even when the images are unrelated, suggesting that trajectories in image space and weight space are structurally aligned.

## 5 CONCLUSION

In this work, we examined the advantages and disadvantages of different initialization strategies for INRs and proposed PCAInit , a training-free initialization which achieves state-of-the-art results on image reconstruction with lower training time. We have shown that First Frame initialization can serve as a practical alternative to Meta in the context of video frames. Although Previous Frame is not an efficient strategy for video frame reconstruction, its gradual frame-to-frame improvement suggests potential as a regularization mechanism to enforce temporal consistency within sequences.

Overall, our empirical results demonstrate that PCAInit is broadly effective, performing well not only on video frames but also on unrelated images. While we are still working toward a complete theoretical explanation, our findings suggest that INRs, and SIREN in particular, if capable of rep-

resenting images accurately, also learn to capture structure in compressed PCA (latent) spaces. A related observation appears in the "Learning a Space of Implicit Functions" section of the SIREN paper (Sitzmann et al., 2020b), where a hypernetwork $\Psi$ maps a latent code vector $z$ to the weights of a SIREN in order to reconstruct signals from sparse pixel observations. Sitzmann et al. (2020b) conclude that generalization over SIREN representations is at least as powerful as generalization over images. This suggests that the set of SIREN weights that represent natural images forms a structured, low-dimensional manifold that is closely related to the manifold of images itself. In our setting, each frame is well represented by a SIREN with a fixed architecture, and optimization moves the weights within a low-dimensional subset of parameter space that corresponds to the observed variability in the images. When we apply PCA to both the image space and the corresponding weight space, PCA extracts the main directions along which frames vary. Since the same underlying content changes drive both the pixel values and the optimized weights, it is natural that the principal components in the two spaces align up to an approximately orthogonal transform and a linear projection, which is precisely what we observe empirically. This alignment between image space and weight space may underlie the robustness of PCAInit, though at present engineering practice is several steps ahead of theoretical understanding. As others have remarked in related work (Shazeer, 2020), sometimes architectures succeed for reasons that remain elusive, and progress is driven by empirical discovery before theory catches up.

For future work, we will investigate whether the observed alignment between image space and weight space also holds for other INR architectures, such as WIRE (Saragadam et al., 2023), Finer (Liu et al., 2024), NeRV (Chen et al., 2021), and NIRVANA (Maiya et al., 2023). If this phenomenon proves to be robust across architectures and datasets, it could be exploited as an additional evaluation metric or even as a regularization term in the loss function. Moreover, if we can directly predict $\mathrm{flat}(\theta_t^*)$ from $\mathbf{V}_{\mathrm{PCA}}$ via this alignment (recall Section 3.3), we would obtain a (potentially training-free) mapping from images or video frames to MLP parameters. Such a mapping would provide compact implicit representations of visual data and open up a promising direction for INR-based image and video compression.

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

# A APPENDIX

## A.1 DETAILED EXPERIMENTAL SETUPS

Table 3: Selected videos from DAVIS 2017 at 480p resolution. In total, 15 videos with 991 frames are used.

| Camera Type | Video | #Frames | Size | Notes |
|---|---|---|---|---|
| | bike-packing | 69 | **(480, 910, 3)** | Different resolution |
| Fixed view | breakdance-flare | 71 | | – |
| | crossing | 52 | | – |
| | dog-agility | 25 | (480, 854, 3) | Fast movement |
| | drift-chicane | 52 | | Smoke |
| | planes-water | 38 | | Blue |
| Follow object | bear | 82 | | – |
| | boxing-fisheye | 87 | | Distortion |
| | dog-gooses | 86 | | Green |
| | gold-fish | 78 | (480, 854, 3) | Blue |
| | parkour | 100 | | Fast movement |
| | snowboard | 66 | | White, fast movement |
| | tennis | 70 | | Fast movement |
| Zoom-in/out | kid-football | 68 | (480, 854, 3) | Zoom-out |
| | lab-coat | 47 | | Zoom-in then zoom-out |

## A.2 PCA ON THE DAVIS 2017 VIDEO AT 480P RESOLUTION

We show our analysis of video frames and optimized MLPs on 15 selected videos in DAVIS 2017 dataset (Pont-Tuset et al., 2017).

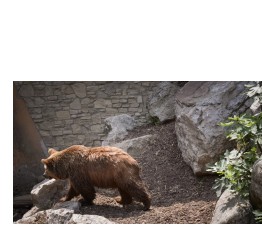 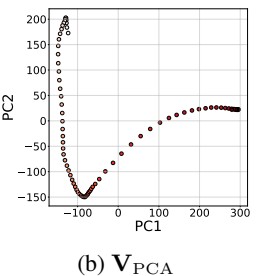 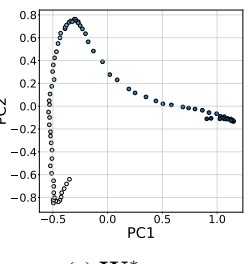 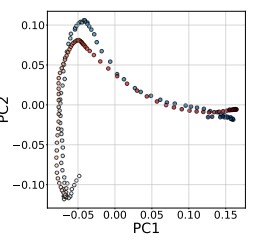

(a) Video frame  (b) $\mathbf{V}_{\mathrm{PCA}}$  (c) $\mathbf{W}^*_{\mathrm{PCA}}$  (d) rot($\mathbf{V}_{\mathrm{PCA}}$) & $\mathbf{W}^*_{\mathrm{PCA}}$

Figure 6: bear (Procrustes disparity: 0.021)

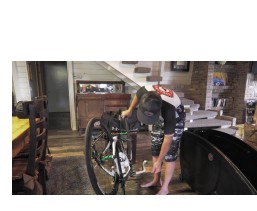 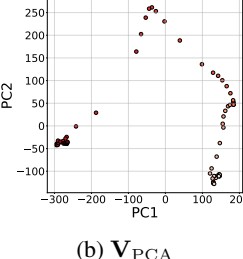 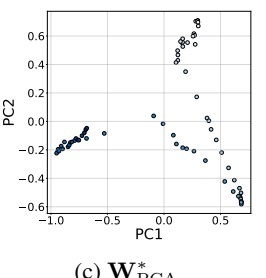 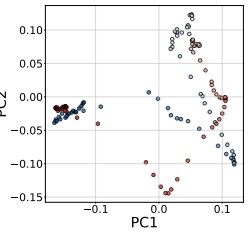

(a) Video frame  (b) $\mathbf{V}_{\mathrm{PCA}}$  (c) $\mathbf{W}^*_{\mathrm{PCA}}$  (d) rot($\mathbf{V}_{\mathrm{PCA}}$) & $\mathbf{W}^*_{\mathrm{PCA}}$

Figure 7: bike-packing (Procrustes disparity: 0.203)

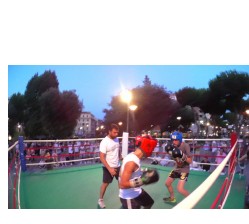 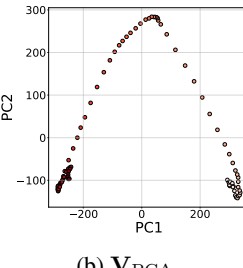 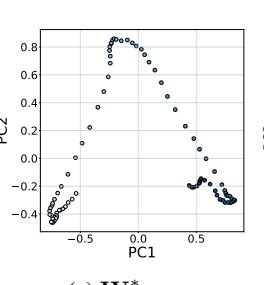 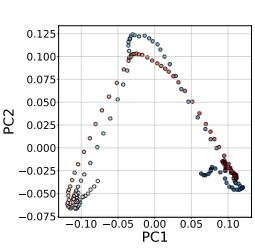

(a) Video frame  (b) $\mathbf{V}_{\mathrm{PCA}}$  (c) $\mathbf{W}^*_{\mathrm{PCA}}$  (d) rot($\mathbf{V}_{\mathrm{PCA}}$) & $\mathbf{W}^*_{\mathrm{PCA}}$

Figure 8: boxing-fisheye (Procrustes disparity: 0.025)

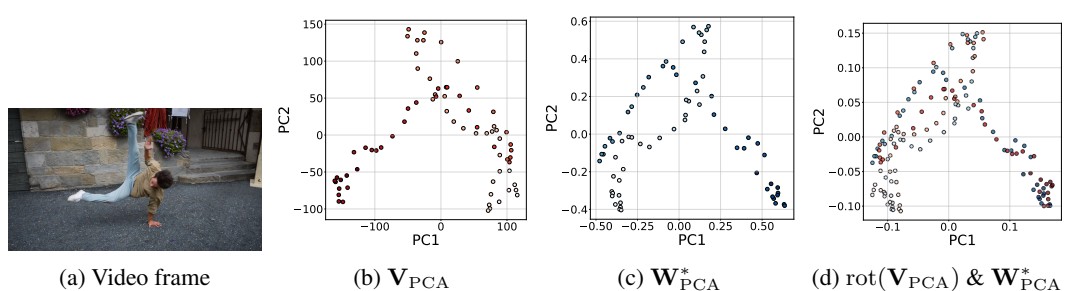

Figure 9: breakdance-flare (Procrustes disparity: 0.024)

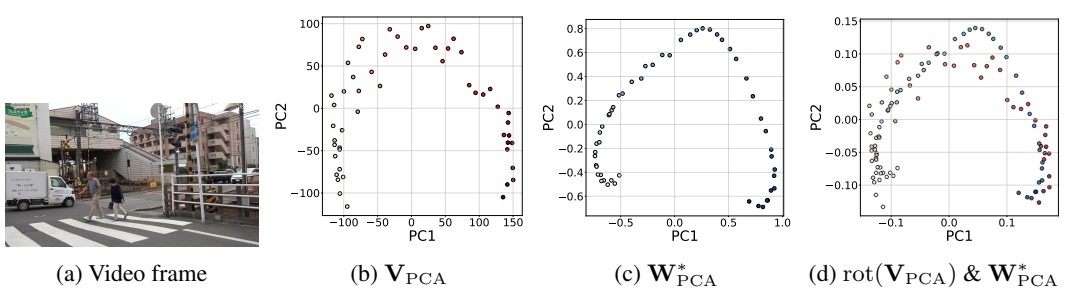

Figure 10: crossing (Procrustes disparity: 0.077)

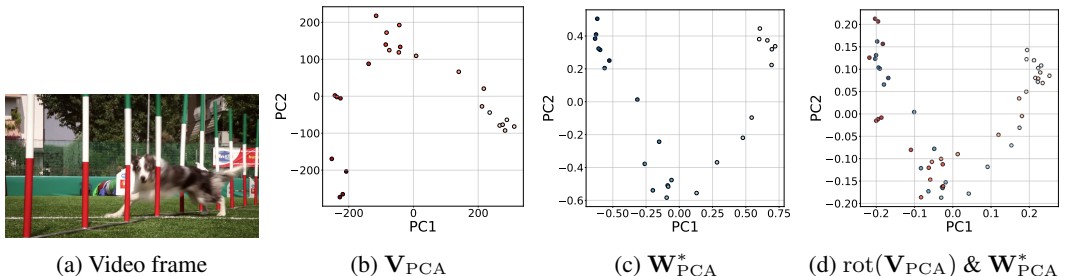

Figure 11: dog-agility (Procrustes disparity: 0.107)

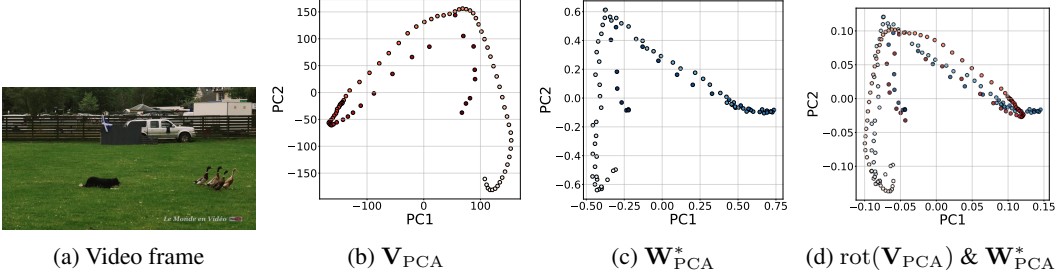

Figure 12: dog-gooses (Procrustes disparity: 0.021)

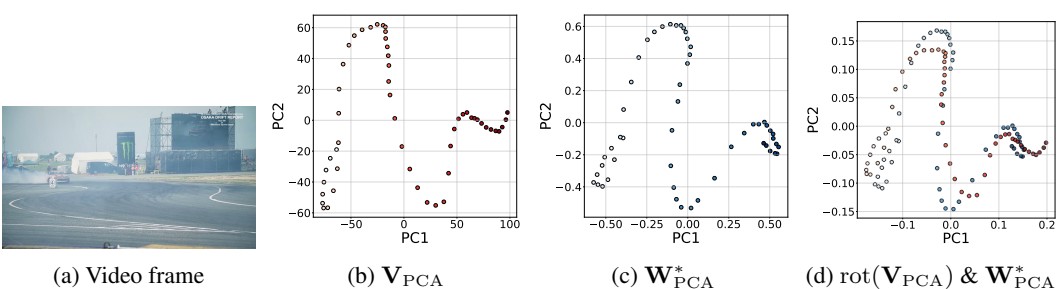

(a) Video frame      (b) $\mathbf{V}_{\mathrm{PCA}}$      (c) $\mathbf{W}^*_{\mathrm{PCA}}$      (d) $\mathrm{rot}(\mathbf{V}_{\mathrm{PCA}})$ & $\mathbf{W}^*_{\mathrm{PCA}}$

Figure 13: drift-chicane (Procrustes disparity: 0.053)

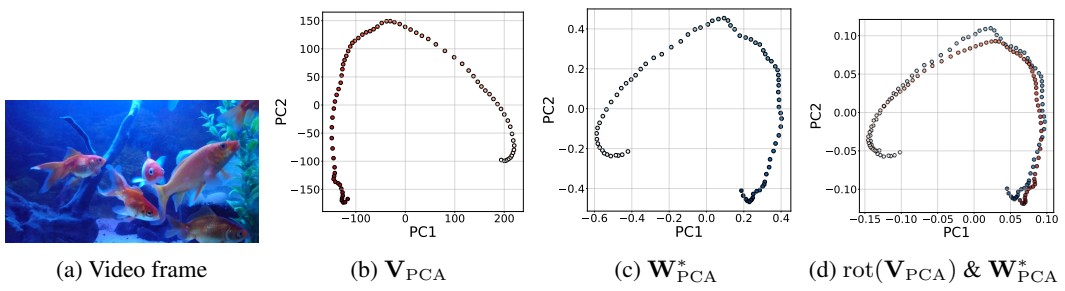

(a) Video frame      (b) $\mathbf{V}_{\mathrm{PCA}}$      (c) $\mathbf{W}^*_{\mathrm{PCA}}$      (d) $\mathrm{rot}(\mathbf{V}_{\mathrm{PCA}})$ & $\mathbf{W}^*_{\mathrm{PCA}}$

Figure 14: gold-fish (Procrustes disparity: 0.012)

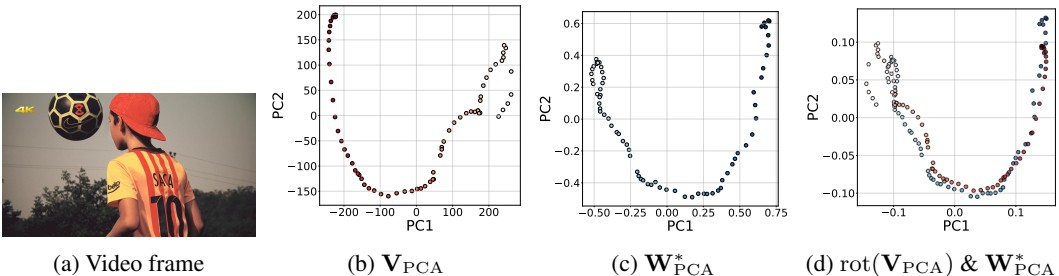

(a) Video frame      (b) $\mathbf{V}_{\mathrm{PCA}}$      (c) $\mathbf{W}^*_{\mathrm{PCA}}$      (d) $\mathrm{rot}(\mathbf{V}_{\mathrm{PCA}})$ & $\mathbf{W}^*_{\mathrm{PCA}}$

Figure 15: kid-football (Procrustes disparity: 0.024)

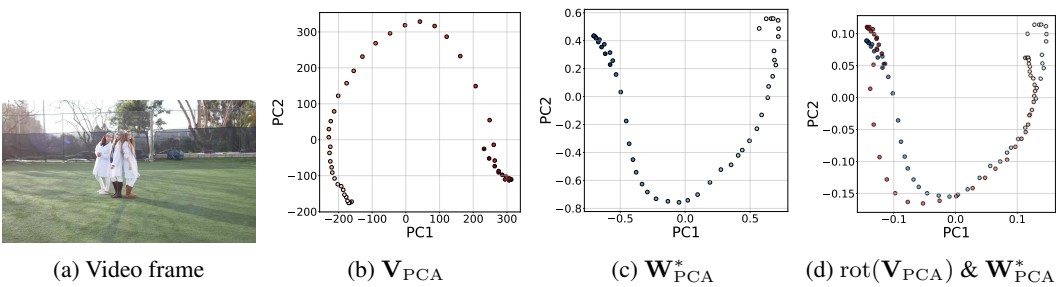

(a) Video frame      (b) $\mathbf{V}_{\mathrm{PCA}}$      (c) $\mathbf{W}^*_{\mathrm{PCA}}$      (d) $\mathrm{rot}(\mathbf{V}_{\mathrm{PCA}})$ & $\mathbf{W}^*_{\mathrm{PCA}}$

Figure 16: lab-coat (Procrustes disparity: 0.063)

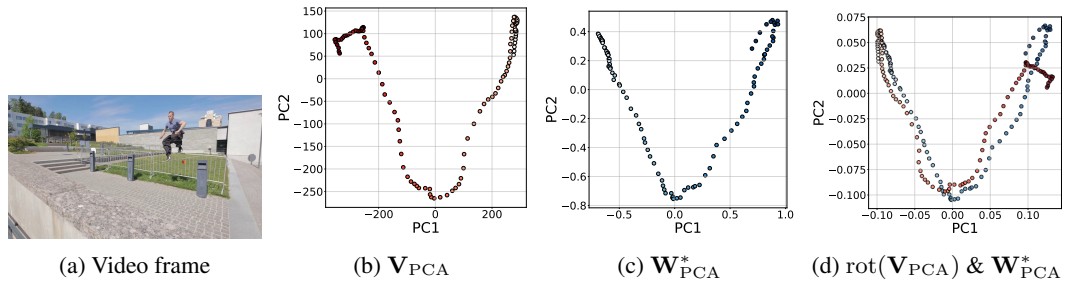

(a) Video frame     (b) $\mathbf{V}_{\text{PCA}}$     (c) $\mathbf{W}^*_{\text{PCA}}$     (d) rot($\mathbf{V}_{\text{PCA}}$) & $\mathbf{W}^*_{\text{PCA}}$

Figure 17: parkour (Procrustes disparity: 0.078)

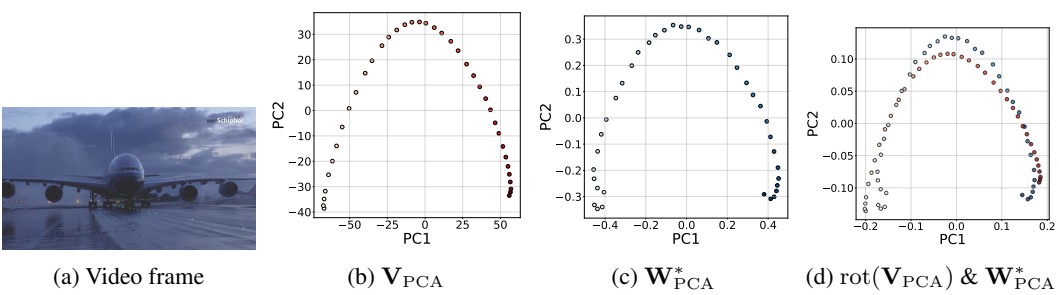

(a) Video frame     (b) $\mathbf{V}_{\text{PCA}}$     (c) $\mathbf{W}^*_{\text{PCA}}$     (d) rot($\mathbf{V}_{\text{PCA}}$) & $\mathbf{W}^*_{\text{PCA}}$

Figure 18: planes-water (Procrustes disparity: 0.026)

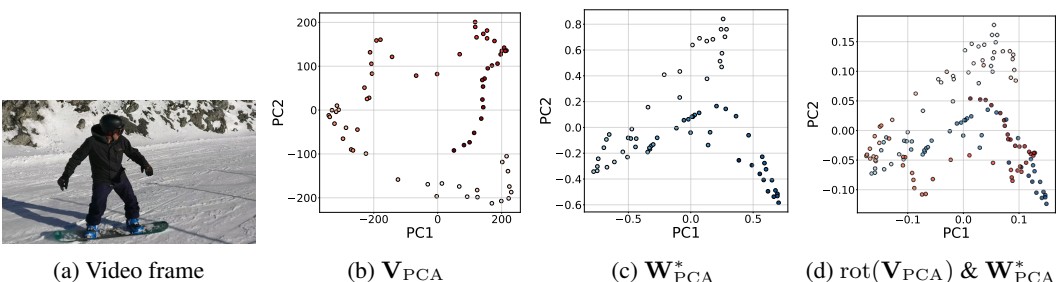

(a) Video frame     (b) $\mathbf{V}_{\text{PCA}}$     (c) $\mathbf{W}^*_{\text{PCA}}$     (d) rot($\mathbf{V}_{\text{PCA}}$) & $\mathbf{W}^*_{\text{PCA}}$

Figure 19: snowboard (Procrustes disparity: 0.400)

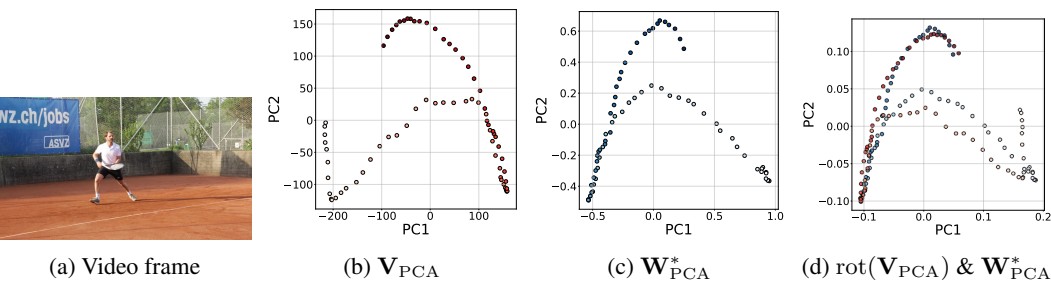

(a) Video frame     (b) $\mathbf{V}_{\text{PCA}}$     (c) $\mathbf{W}^*_{\text{PCA}}$     (d) rot($\mathbf{V}_{\text{PCA}}$) & $\mathbf{W}^*_{\text{PCA}}$

Figure 20: tennis (Procrustes disparity: 0.056)

### A.3 FURTHER EXPERIMENTS

We performed additional experiments and the results are reported as mean and standard deviation of PSNR and SSIM across frames on *bear* video at 480p resolution. The highest scores are shown in **bold**.

#### A.3.1 PCA ON GRAYSCALE ($d = 2$) AND RGB ($d = 3$)

Table 4: Comparison of PCAInit results when PCA is applied with different input modalities and dimensions. Both RGB ($d = 2$) and Gray ($d = 2$) achieve very similar performance, while RGB ($d = 3$) gives slightly higher PSNR and SSIM. For simplicity and consistency, we use RGB with $d = 2$ in all main experiments.

| Method | PSNR (dB) ↑ | SSIM ↑ |
|---|---|---|
| PCAInit (RGB, $d = 2$) | $44.49 \pm 0.42$ | $0.992 \pm 0.001$ |
| PCAInit (Gray, $d = 2$) | $44.53 \pm 0.32$ | $0.992 \pm 0.001$ |
| PCAInit (RGB, $d = 3$) | $\mathbf{44.65 \pm 0.44}$ | $\mathbf{0.993 \pm 0.001}$ |

#### A.3.2 NOISE RANGE

Table 5: Ablation study on different uniform noise ranges for PCAInit on the *bear* video. Noise is sampled from $\mathcal{U}[-r, r)$ using `numpy.random.uniform`, where $r$ denotes the noise range. Although $r = 10^{-2}$ (bold) achieved the highest PSNR, the performance across $10^{-5}$, $10^{-4}$, and $10^{-3}$ was nearly identical. We therefore chose $r = 10^{-3}$ for our experiments, as it represents a stable middle ground between the smaller and larger ranges.

| Noise Range $r$ | PSNR (dB) ↑ | SSIM ↑ |
|---|---|---|
| $10^{-5}$ | $44.48 \pm 0.37$ | $0.992 \pm 0.001$ |
| $10^{-4}$ | $44.46 \pm 0.35$ | $0.992 \pm 0.001$ |
| $10^{-3}$ | $44.45 \pm 0.39$ | $0.992 \pm 0.001$ |
| $10^{-2}$ | $\mathbf{44.74 \pm 0.39}$ | $\mathbf{0.993 \pm 0.001}$ |
| $10^{-1}$ | $30.13 \pm 0.15$ | $0.828 \pm 0.008$ |

#### A.3.3 MLP ARCHITECTURES

Table 6: Ablation study on different MLP architectures on SIREN and PCAInit methods. The setting with 512 hidden units and 3 hidden layers achieves the best results and is used throughout the paper.

| # Hidden | | SIREN | | PCAInit | |
|---|---|---|---|---|---|
| Units | Layers | PSNR ↑ | SSIM ↑ | PSNR ↑ | SSIM ↑ |
| 128 | 2 | $23.32 \pm 0.46$ | $0.547 \pm 0.015$ | $24.99 \pm 0.49$ | $0.673 \pm 0.016$ |
| | 3 | $24.83 \pm 0.55$ | $0.664 \pm 0.017$ | $26.38 \pm 0.76$ | $0.728 \pm 0.031$ |
| 256 | 2 | $25.72 \pm 0.48$ | $0.739 \pm 0.014$ | $29.58 \pm 0.46$ | $0.883 \pm 0.005$ |
| | 3 | $27.25 \pm 0.35$ | $0.804 \pm 0.013$ | $33.82 \pm 0.40$ | $0.940 \pm 0.003$ |
| 512 | 2 | $28.15 \pm 0.49$ | $0.843 \pm 0.014$ | $35.48 \pm 0.43$ | $0.964 \pm 0.003$ |
| | 3 | $\mathbf{32.46 \pm 0.73}$ | $\mathbf{0.923 \pm 0.024}$ | $\mathbf{44.49 \pm 0.42}$ | $\mathbf{0.992 \pm 0.001}$ |

## A.4 ZERO-SHOT SUPER-RESOLUTION

This experiment evaluates high-resolution generalization, i.e., whether models trained on low-resolution frames can be directly applied to higher-resolution inputs without retraining. Since the DAVIS 2017 dataset (Pont-Tuset et al., 2017) also provides full-HD resolution, we reuse the optimized MLPs from Section 4.1, trained on the *bear* video at 480p, to infer full-HD frames of size $(1080, 1920)$ and compare them against the ground-truth full-HD *bear* video.

From Table 7, PCAInit attains the highest PSNR (29.83 dB), while also achieving a SSIM score (0.853) that is nearly identical to Meta (0.854). These results indicate that PCAInit generalizes effectively to high-resolution inputs without retraining, offering reconstruction quality comparable to meta-learned initialization but without the associated training overhead.

Table 7: Mean and standard deviation of PSNR and SSIM results for the zero-shot super-resolution task on the *bear* video. Best results are shown in **bold**. PCAInit achieves the highest PSNR and nearly matches the best SSIM, demonstrating strong generalization to high-resolution frames without retraining.

| Method | PSNR (dB) ↑ | SSIM ↑ |
|---|---|---|
| $\text{FF}_{PE}$ | $11.31 \pm 0.39$ | $0.071 \pm 0.007$ |
| $\text{NeRF}_{PE}$ | $20.96 \pm 0.49$ | $0.325 \pm 0.020$ |
| SIREN | $28.62 \pm 0.53$ | $0.817 \pm 0.021$ |
| Meta | $29.53 \pm 0.62$ | $\mathbf{0.854 \pm 0.016}$ |
| Prev Frame | $28.57 \pm 0.75$ | $0.798 \pm 0.030$ |
| First Frame | $29.10 \pm 0.65$ | $0.838 \pm 0.015$ |
| PCAInit | $\mathbf{29.83 \pm 0.60}$ | $0.853 \pm 0.010$ |

## A.5 PCAInit PER EPOCH

Recall Figure 3, where we plot the mean and standard deviation of PSNR across frames per epoch on the *bear* video. Here, we plot the mean PSNR of PCAInit over all 15 DAVIS (Pont-Tuset et al., 2017) videos to illustrate its convergence behavior.

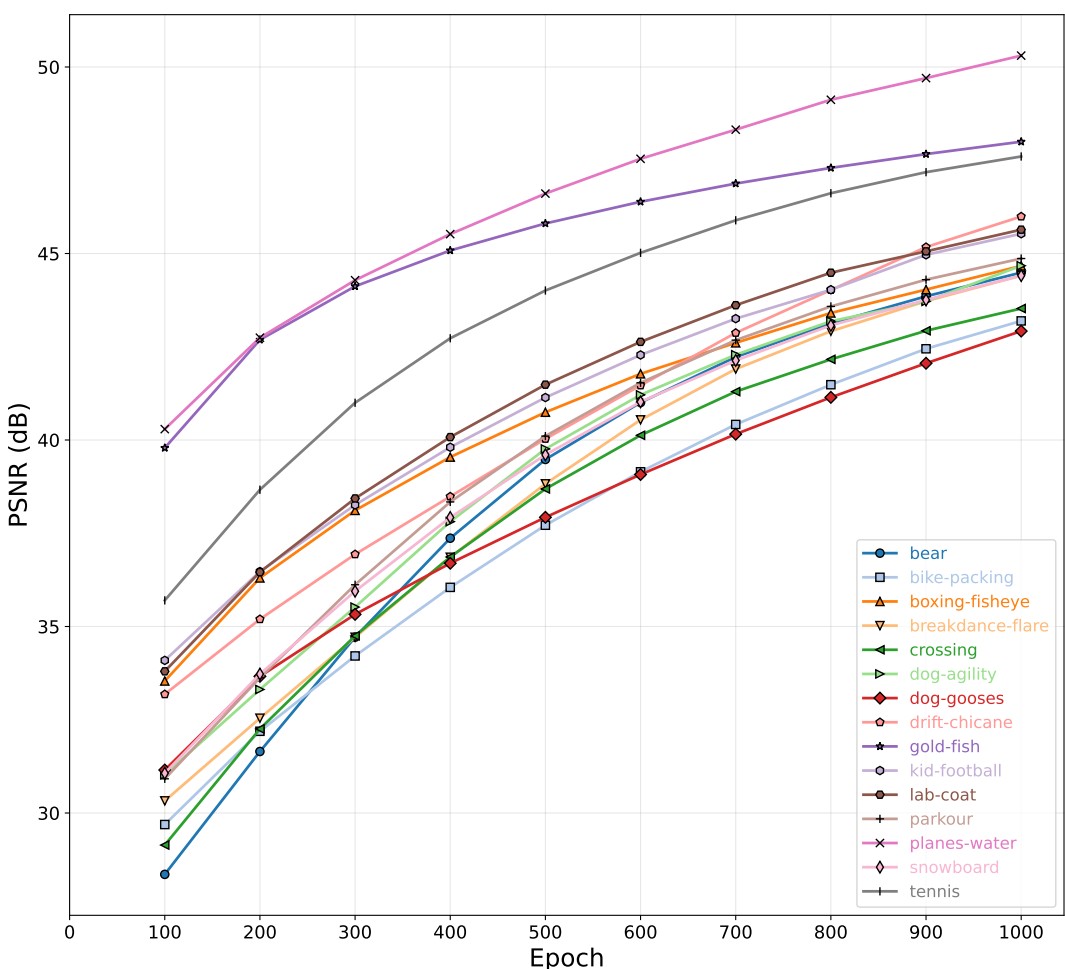

Figure 21: Mean PSNR across frames per epoch for PCAInit on the 15 DAVIS videos.

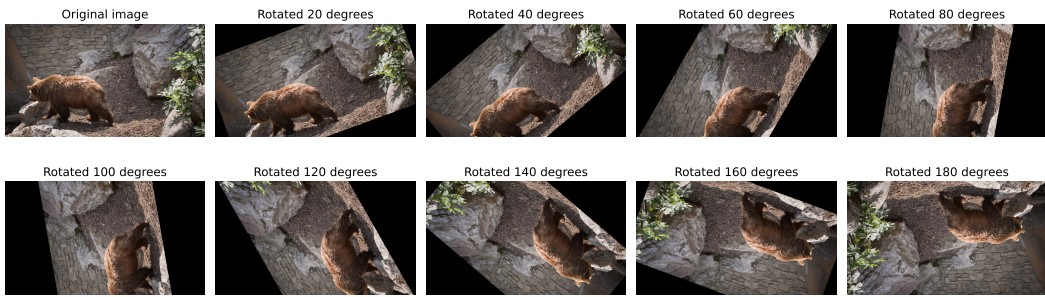

Figure 22: From the original image, we create nine rotated images. These rotated images serve as additional inputs for running PCAInit .

## A.6    PCAInit on a Single Image

Recall Section 3.5, our PCAInit requires more than one image to operate. Therefore, to apply it to a single image, we naively create a set of rotated variants of the original image, as shown in Figure 22.

We then test PCAInit on collections consisting of the original image and different numbers of rotated images, which act as additional images for running PCA, and we report the results in Table 8. For example, "Original + 1" means that the image stack contains the original image and one rotated image, where the rotated image is rotated by 20 degrees. The same applies to "Original + 4" and "Original + 9", which follow the order in Figure 22, with "Original + 9" including all images.

Table 8: We report the mean and standard deviation of PSNR of PCAInit for each collection. We also report the PSNR of the original image in each collection and the PSNR of SIREN on the original image for comparison. Best results are shown in **bold**.

|  | **Original + 1** | **Original + 4** | **Original + 9** |
|---|---|---|---|
| Collection | **44.85 $\pm$ 0.01** | 43.97 $\pm$ 0.27 | 44.19 $\pm$ 0.54 |
| Original image | **44.86** | 44.22 | 44.41 |
| SIREN |  | 32.64 |  |

The results in Table 8 show that our PCAInit works well with additional images. Surprisingly, "Original + 1" achieves the highest PSNR, with a +12.22 dB improvement compared to SIREN, which means that our PCAInit only needs one additional image to operate well.

## A.7 REMOVING SIREN INITIALIZATION IN PCAINIT

Recall Section 3.4, where we introduced PCAInit . Here, we perform an ablation study by removing the SIREN initialization and using the default initialization of the PyTorch `Linear` layer, where the weights are sampled from a uniform distribution $U(-\text{bound}, \text{bound})$, with bound $= 1/\sqrt{\text{in\_features}}$. In this setting, $\mathbf{W}^{\text{SIREN}}$ is replaced by $\mathbf{W}^{\text{PyTorch}}$.

The results in Table 9 show that, without SIREN initialization, the model fails to converge. This behavior is consistent with the findings of Sitzmann et al. (2020b), where they demonstrate that this specific initialization scheme is necessary for training SIREN.

Table 9: Mean and standard deviation of PSNR and SSIM results on the *bear* video when using default PyTorch initialization versus SIREN initialization in PCAInit . The highest scores are shown in **bold**.

| Initialization | PSNR (dB) ↑ | SSIM ↑ |
|---|---|---|
| $\mathbf{W}^{\text{PyTorch}}$ | $13.05 \pm 0.22$ | $0.078 \pm 0.009$ |
| $\mathbf{W}^{\text{SIREN}}$ | $\mathbf{44.49 \pm 0.42}$ | $\mathbf{0.992 \pm 0.001}$ |

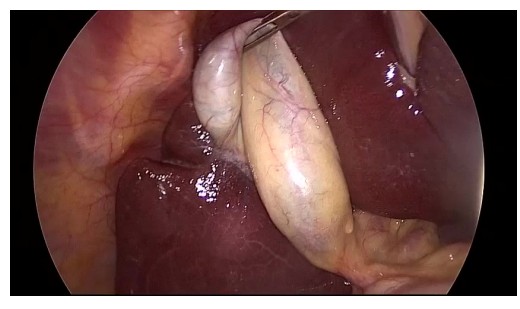 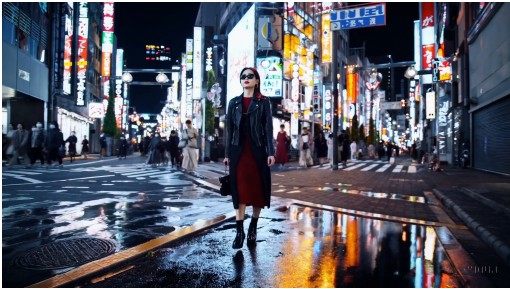

(a) Cholec80 (*video01*)                    (b) *tokyo-walk*

Figure 23: First frame of (a) *video01* from the Cholec80 surgical dataset and (b) the *tokyo-walk* video generated by SORA.

## A.8    PCAINIT ON NON-NATURAL & AI-GENERATED VIDEOS

We conduct additional experiments on two new types of videos (as shown in Figure 23):

**Non-natural video.** We use the *video01* sequence from the Cholec80 (Twinanda et al., 2016) surgical video dataset. For simplicity, we refer to the *video01* sequence as *Cholec80* in the following experiments on non-natural videos. We extract frames at 1 fps and use the first 82 frames to reduce training time. Each frame has a resolution of $(480, 854)$.

**AI-generated video.** We use the *tokyo-walk* video generated by SORA (OpenAI)[3]. We extract 60 frames at 1 fps with an original resolution of $(1080, 1920)$, and then resize them to $(480, 854)$ to reduce training time and to match the resolution of the other videos described in Section A.1.

In Table 10, for the Cholec80 results, although PCAInit has a lower PSNR (43.57 dB) than Prev Frame (45.45 dB), it achieves the lowest standard deviation, indicating more consistent performance across frames (a behavior also observed in the experiments in Section 4). For the *tokyo-walk* results, PCAInit is the best method, achieving the highest PSNR (43.57 dB). Qualitative reconstructions are shown in Figure 24 and Figure 25 for Cholec80 and *tokyo-walk*, respectively.

In conclusion, these results confirm that PCAInit is effective not only on natural videos, but also on surgical as well as AI-generated videos.

Table 10: Mean and standard deviation of PSNR and SSIM results on Cholec80 and *tokyo-walk* when using different initialization methods. Best results in **bold**, second-best underlined.

| Method | Cholec80 | | *tokyo-walk* | |
|---|---|---|---|---|
| | **PSNR** (dB) ↑ | **SSIM** ↑ | **PSNR** (dB) ↑ | **SSIM** ↑ |
| SIREN | $34.06 \pm 1.14$ | $0.928 \pm 0.009$ | $35.59 \pm 3.11$ | $0.961 \pm 0.014$ |
| Meta | $39.16 \pm 2.40$ | $0.963 \pm 0.062$ | $38.11 \pm 4.16$ | $0.968 \pm 0.033$ |
| Prev Frame | $\mathbf{45.45 \pm 3.53}$ | $\mathbf{0.987 \pm 0.012}$ | $\underline{41.51 \pm 3.70}$ | $0.979 \pm 0.009$ |
| First Frame | $37.54 \pm 1.96$ | $0.959 \pm 0.024$ | $39.08 \pm 4.01$ | $0.973 \pm 0.025$ |
| PCAInit | $\underline{43.57 \pm 0.65}$ | $\mathbf{0.987 \pm 0.002}$ | $\mathbf{43.98 \pm 2.58}$ | $\mathbf{0.991 \pm 0.003}$ |

---

[3]https://cdn.openai.com/sora/videos/tokyo-walk.mp4

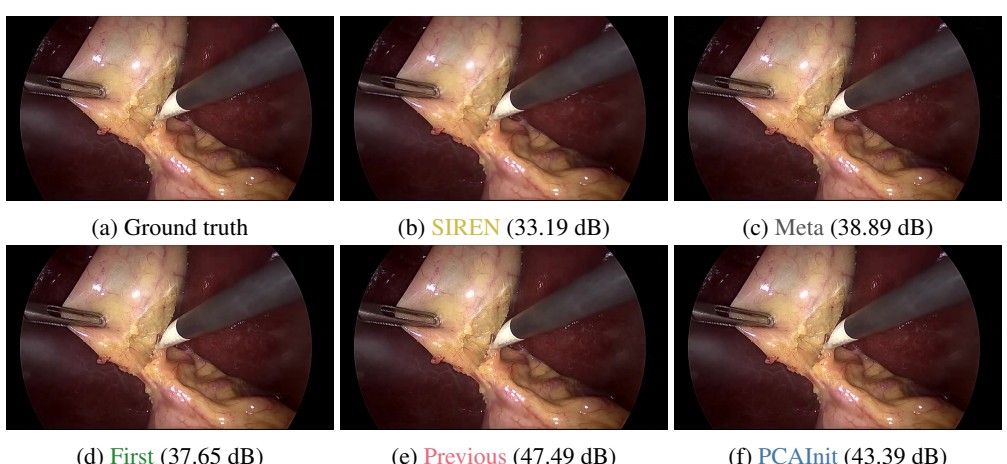

(a) Ground truth     (b) SIREN (33.19 dB)     (c) Meta (38.89 dB)

(d) First (37.65 dB)     (e) Previous (47.49 dB)     (f) PCAInit (43.39 dB)

Figure 24: Reconstruction of the ground-truth last frame ($82^{nd}$) of the Cholec80 video using different methods in Section A.8.

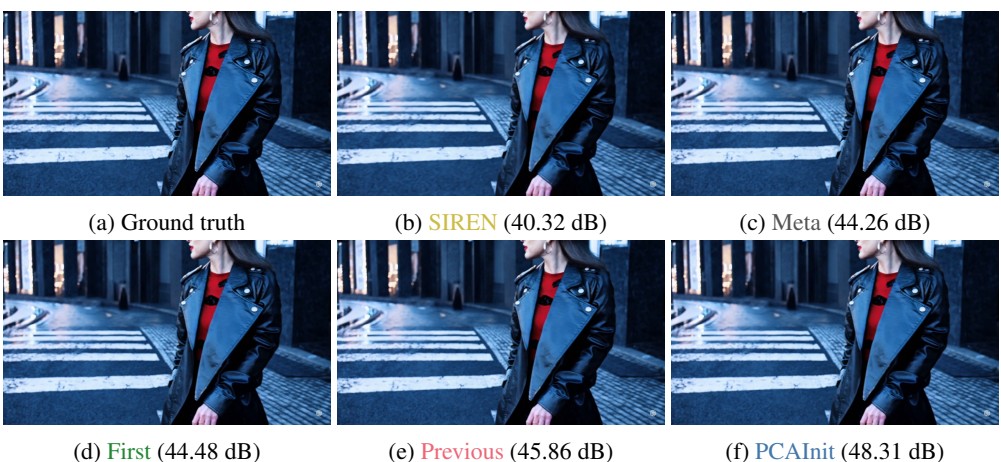

(a) Ground truth     (b) SIREN (40.32 dB)     (c) Meta (44.26 dB)

(d) First (44.48 dB)     (e) Previous (45.86 dB)     (f) PCAInit (48.31 dB)

Figure 25: Reconstruction of the ground-truth last frame ($60^{th}$) of the *tokyo-walk* video using different methods in Section A.8.

## A.9 CROSS-VIDEO INITIALIZATION WITH PCAINIT

Recall Section 3.4, where we introduced PCAInit to construct the weight trajectory $\mathbf{W}^{\mathbf{V}}$. Here, we perform cross-video initialization by first applying PCAInit on the *bear* video ($T$=82) to obtain

$$\mathbf{W}^{\mathbf{V}_{\text{bear}}} = \{\text{flat}(\theta_t^{\mathbf{V}_{\text{bear}}})\}_{t=1}^{82} \in \mathbb{R}^{82 \times n_\theta},$$

where $\theta_t^{\mathbf{V}_{\text{bear}}}$ denotes the initialization for frame $t$ derived from the *bear* video $\mathbf{V}_{\text{bear}}$. Each $\theta_t^{\mathbf{V}_{\text{bear}}}$ is then used to initialize the corresponding $\theta_t$ of other videos, instead of recomputing PCAInit for each of them.

We evaluate this cross-video initialization on three videos:

- $\{\text{flat}(\theta_t^{\mathbf{V}_{\text{bear}}})\}_{t=1}^{69}$ for the *bike-packing* video ($T$=69) from the DAVIS 2017 dataset,

- $\{\text{flat}(\theta_t^{\mathbf{V}_{\text{bear}}})\}_{t=1}^{82}$ for the Cholec80 video ($T$=82) introduced in Section A.8,

- $\{\text{flat}(\theta_t^{\mathbf{V}_{\text{bear}}})\}_{t=1}^{60}$ for the *tokyo-walk* video ($T$=60) introduced in Section A.8.

The results in Table 11 show that initialization with $\mathbf{W}^{\mathbf{V}_{\text{bear}}}$ yields performance comparable to running PCAInit on each video individually, indicating that these weights transfer well across videos. This suggests that we can compute PCAInit once on the longest video $\mathbf{V}_{T_{\max}}$ with $T_{\max}$ frames, and reuse $\{\text{flat}(\theta_t^{\mathbf{V}_{T_{\max}}})\}_{t=1}^{T_{\max}}$ for other videos, reducing both computation time and memory usage, since we no longer need to store a separate $\mathbf{W}^{\mathbf{V}}$ for each video.

Table 11: Mean and standard deviation of PSNR and SSIM when reusing the *bear* PCAInit weights $\mathbf{W}^{\mathbf{V}_{\text{bear}}}$ to initialize other videos, compared to using PCAInit computed separately on each video. Cross-video initialization from *bear* achieves performance on par with running PCAInit on each video individually.

| Video | Initialization | PSNR (dB) ↑ | SSIM ↑ |
|---|---|---|---|
| *bike-packing* | $\mathbf{W}^{\mathbf{V}_{\text{bear}}}$ | $43.14 \pm 0.52$ | $0.989 \pm 0.001$ |
| | $\mathbf{W}^{\mathbf{V}_{\text{bike-packing}}}$ | $43.19 \pm 0.46$ | $0.989 \pm 0.001$ |
| Cholec80 | $\mathbf{W}^{\mathbf{V}_{\text{bear}}}$ | $43.78 \pm 0.70$ | $0.987 \pm 0.001$ |
| | $\mathbf{W}^{\mathbf{V}_{\text{Cholec80}}}$ | $43.57 \pm 0.65$ | $0.987 \pm 0.002$ |
| *tokyo-walk* | $\mathbf{W}^{\mathbf{V}_{\text{bear}}}$ | $43.79 \pm 2.70$ | $0.990 \pm 0.004$ |
| | $\mathbf{W}^{\mathbf{V}_{\text{tokyo-walk}}}$ | $43.98 \pm 2.58$ | $0.991 \pm 0.003$ |

