# OpenReview forum: "PCAInit: Training-Free Initialization for Image-Based Neural Representations"
_ICLR.cc/2026/Conference — Submitted to ICLR 2026_

### Official Review · Reviewer_pSrS · 2025-10-24

**Soundness:** 4
**Presentation:** 3
**Contribution:** 4
**Rating:** 8
**Confidence:** 4

**Summary:**

The paper proposes a method for INR initialization. The method uses PCA to align the initial weight trajectory over time to that of video frames over time. This can also be applied to a collection of images in the same way. The method shows improved reconstruction accuracy over the naive initialization of several prior works.

**Strengths:**

- The method is a simple, actionable, method for improving INR performance from better initialization

**Weaknesses:**

- Unclear if this affects bitrate
- More comparisons would be nice

**Questions:**

This paper provides a simple framework for improving INR results by using a better initialization. Overall the method is simple, straightforward to implement, and achieves clear improvements in results. It's the kind of thing that I can imagine becoming standard in the INR world. I do have some minor issues which I think should be clarified though:

1. How does the method perform on non-SIREN networks? I think the only evaluation was on SIREN but it could be applied to other methods as well. I expected to see a table comparing for several methods: "Naive init" vs "Our init".
2. I think there should be some comparison to NIRVANA because they did some work on the initialization scheme as well

---

> ### Author Response · Authors · 2025-11-20
> **Response to Reviewer pSrS**
>
> > Q1: Unclear if this affects bitrate
>
> A1: In this paper, we use video frame fitting mainly as a demonstration of the initialization approaches and focus on reconstruction quality (for example PSNR and SSIM). We do not measure bitrate, so we cannot quantify the impact on bitrate in the current work and leave this as future work.
>
> ---
> > Q2: More comparisons would be nice
>
> A2: **Additional experiments.** In line with the reviewer's comments, we have conducted several additional comparisons and experiments. These include new baselines, experiments on image initialization transferability, as well as experiments on medical videos and AI-generated video clips. We have included these results in the revised version of the paper as well as in the revised appendix.
>
> ---
> > Q3: How does the method perform on non-SIREN networks? I think the only evaluation was on SIREN but it could be applied to other methods as well. I expected to see a table comparing for several methods: "Naive init" vs "Our init".
>
> A3: We thank the reviewer for this question. In the current work, we focus on SIREN as the base INR and leave a systematic evaluation on other architectures as future work.
>
> **Additional experiments.** This is an important point and we would like to thank the reviewer for bringing this to our attention. In the revised manuscript, we added additional experiments where we use "Naive init" and compare it to our proposed initialization. These results can be found in Appendix A.7 "Removing SIREN initialization in PCAInit". We observe that default initialization yields very poor reconstruction (PSNR 13.05 dB, SSIM 0.078), whereas SIREN initialization reaches 44.49 dB PSNR and 0.992 SSIM (please see Table 9 in the revised paper for detailed results).
>
> ---
> > Q4: I think there should be some comparison to NIRVANA because they did some work on the initialization scheme as well.
>
> A4: We thank the reviewer for this comment. It is true that our Previous Frame initialization is related to NIRVANA. However, NIRVANA fits SIREN in a feature space, and the resulting features are then fed to a NeRV-like block to produce RGB values, while we fit SIREN directly in the image space.
>
> We would like to emphasize that our goal is not to propose a new video representation method, but to create a clean per-frame SIREN setting to study the alignment between image and weight spaces and to design initialization strategies for INRs. In the revised version, we will clarify the conceptual relationship to NIRVANA and explain that a full quantitative comparison would require a different experimental setup, which we leave for future work.

---

### Official Review · Reviewer_QzFL · 2025-10-28

**Soundness:** 3
**Presentation:** 2
**Contribution:** 2
**Rating:** 4
**Confidence:** 5

**Summary:**

The paper focuses on improving implicit neural representations (INRs) for video data, which are often limited by high training costs and suboptimal reconstruction quality. Using SIREN as the baseline, the authors explore three initialization strategies to accelerate INR training and enhance performance on large-scale video frames. Among them, the proposed PCAInit method is a training-free initialization derived from analyzing the relationship between video frame space and network weight space using principal component analysis (PCA)

**Strengths:**

1. The discovery of a similar PCA trajectory between the image space and the weight space is highly insightful and provides a valuable perspective for understanding the internal structure of INRs.

2. Building on this observation, the proposed PCA-based initialization (PCAInit) is conceptually sound and well-motivated, offering a clear rationale for improving INR performance in video reconstruction.

3. The experimental evaluation is comprehensive and convincing, comparing PCAInit with multiple classical and meta-learned initialization methods across diverse datasets, which strongly supports the validity and effectiveness of the proposed approach.

**Weaknesses:**

1. The method description lacks clarity. In lines 281–294, the phrase “use the PCA basis of ... weight space” is conceptually vague. The authors should present the method using clearly defined mathematical symbols and equations rather than verbal explanations. As written, it is difficult to understand the precise transformation steps between image and weight spaces.
2. The abstract and introduction should emphasize the discovery of similar PCA trajectories in image and weight spaces, which I consider to be the paper’s most valuable contribution. In contrast, the Previous Frame and First Frame initialization strategies are relatively minor contributions and do not represent the core insight of the work, yet the current abstract and introduction fail to highlight this key finding.
3. Although the paper defines its problem in the context of video representation, it uses SIREN, a model originally designed for static image representation. This design choice is unconvincing. The authors mention NeRV in the related works but incorrectly state that NeRV “focuses on compression at the cost of reconstruction quality.” In reality, NeRV provides faster and more accurate video representation, with compression as an added benefit. For representing videos, NeRV remains a more suitable and efficient choice than modeling each frame with separate MLPs.
4. The training time comparison in Figure 1 is somewhat misleading. The reported speedup relies heavily on parallelization, effectively trading memory for time rather than reducing total computation. In scenarios without sufficient parallel hardware, the training time would remain long. This limitation further weakens the argument for applying the proposed method to video representation, where efficiency is crucial. That's why NeRV is more suitable for video representation.

Overall, while the paper presents a valuable finding regarding the relationship between image and weight spaces, its practical applicability to video representation is limited due to methodological and architectural choices.

**Questions:**

1. The statement in line 196 is confusing. Given the initialized weights $\theta_t$, it is unclear why the model still needs to initialize $\theta_t$ again. This redundancy suggests unclear notation or an inconsistency in the description of initialization steps.
2. In line 305, for the “image collections” case, can the PCA be computed from a single image only, or does the method require multiple images to form the PCA basis?

---

> ### Author Response · Authors · 2025-11-20
> **Response to Reviewer QzFL - Part I**
>
> > Q1: The method description lacks clarity. In lines 281–294, the phrase "use the PCA basis of ... weight space" is conceptually vague. The authors should present the method using clearly defined mathematical symbols and equations rather than verbal explanations. As written, it is difficult to understand the precise transformation steps between image and weight spaces.
>
> A1: We would like to thank the reviewer for raising this issue. We agree that the current description is not sufficiently clear. In the revised version, we will introduce a more precise mathematical notation, replacing the vague verbal descriptions where necessary and clearly spelling out the transformation steps between image and weight spaces.
>
> ---
> > Q2: The abstract and introduction should emphasize the discovery of similar PCA trajectories in image and weight spaces, which I consider to be the paper’s most valuable contribution. In contrast, the Previous Frame and First Frame initialization strategies are relatively minor contributions and do not represent the core insight of the work, yet the current abstract and introduction fail to highlight this key finding.
>
> A2: We thank the reviewer for this insightful comment. Our intention in the current draft was to describe the discovery process of PCAInit, which started from the Previous Frame and First Frame initialization strategies and then led us to the observation of aligned PCA trajectories in image and weight spaces. As such, we wrote the methods in the abstract in that order.
>
> That said, we agree with the reviewer that the alignment of PCA trajectories is the key insight of the paper. In the revised version, we will update the abstract and introduction to clearly foreground this alignment as the main contribution, and present the Previous Frame and First Frame initialization strategies as supporting elements rather than primary contributions.
>
> ---
> > Q3: Although the paper defines its problem in the context of video representation, it uses SIREN, a model originally designed for static image representation. This design choice is unconvincing. The authors mention NeRV in the related works but incorrectly state that NeRV "focuses on compression at the cost of reconstruction quality". In reality, NeRV provides faster and more accurate video representation, with compression as an added benefit. For representing videos, NeRV remains a more suitable and efficient choice than modeling each frame with separate MLPs.
> > The training time comparison in Figure 1 is somewhat misleading. The reported speedup relies heavily on parallelization, effectively trading memory for time rather than reducing total computation. In scenarios without sufficient parallel hardware, the training time would remain long. This limitation further weakens the argument for applying the proposed method to video representation, where efficiency is crucial. That's why NeRV is more suitable for video representation.
> > Overall, while the paper presents a valuable finding regarding the relationship between image and weight spaces, its practical applicability to video representation is limited due to methodological and architectural choices.
>
> A3: We thank the reviewer for the careful and detailed feedback. We agree that NeRV and related architectures are more suitable and efficient for full video representation than modeling each frame with a separate SIREN, especially when efficiency is the main goal. Our motivation for using per-frame SIRENs was not to propose a new video representation method, but to create a clean setting to study the alignment between image space and weight space and to design initialization strategies for INRs.
>
> We apologize for presenting the work mainly in the context of video representation, which was misleading and caused confusion. The core contribution of our paper is the empirical discovery and use of aligned PCA trajectories in image and weight spaces, which we show on both video frame sequences and image collections. In the revised manuscript, we will refocus the narrative on image representation and initialization for INRs, and present the video frame fitting experiments as a convenient large-scale testbed rather than as a claim that our approach is a competitive video representation method.
>
> We also thank the reviewer for pointing out the inaccurate description of NeRV. We will correct this.

---

> ### Author Response · Authors · 2025-11-20
> **Response to Reviewer QzFL - Part II**
>
> > Q4: The statement in line 196 is confusing. Given the initialized weights $\theta_t$, it is unclear why the model still needs to initialize again. This redundancy suggests unclear notation or an inconsistency in the description of initialization steps.
>
> A4: We thank the reviewer for pointing this out. Indeed, the wording was confusing. In the revised version, we changed "Given initial weights $\theta_t$" to "Given weights $\theta_t$" and clarify that no additional initialization step is performed at this stage.
>
> ---
> > Q5: In line 305, for the "image collections" case, can the PCA be computed from a single image only, or does the method require multiple images to form the PCA basis?
>
> A5: **Additional experiments.** In general, PCAInit requires more than one image to compute a meaningful PCA basis. For the single-image case, we construct a small collection by adding rotated versions of the original image and then run PCAInit on this augmented set. We conducted this experiment and added additional results to the revised appendix, specifically in Section A.6 ("PCAInit on a Single Image"). We find that PCAInit works well in this setting, and the best PSNR is obtained by adding only one additional rotated version of the image.

---

### Official Review · Reviewer_8d9P · 2025-10-31

**Soundness:** 3
**Presentation:** 4
**Contribution:** 3
**Rating:** 6
**Confidence:** 3

**Summary:**

This paper proposes PCAInit, a training free initialization mechanism for INRs. The main
idea is to leverage PCA-based alignment between image space and weight space. This
approach is very insightful and explores the connection between weight space and image
space. I think this is the first work that explores this kind of connection for weight
initialization (haven’t checked the latest works). When it comes to results, it has shown
strong empirical results on the DAVIS dataset.

**Strengths:**

1). Paper is written very well and understandable. Further, the Figure1 helps to understand
what is going on with the approach (overall).
The proposed approach has significant improvement over the baseline.
2). The most interesting observation and the strength is this not only works for videos but also
for unrelated image collections. This is a huge plus.
3). When it comes to novelty, as far as I know, this is a novel approach.

**Weaknesses:**

1). I would like to know why authors specifically selected SIREN as the base architecture? Did authors attempt GAUSS, WIRE or any improved versions of SIREN (for instance FINER).

2). As this paper focuses on improving initialization for video frames, how does PCAInit compare to architectures explicitly designed for videos, such as the NeRV family or related INR models? Can PCAInit also be applied to those networks, or are there
structural limitations?

3). The evaluation is limited to 15 videos from DAVIS 2017, which are all natural videos. It would be valuable to understand how the approach performs on non-natural or AIgenerated videos, or on videos with abrupt scene transitions that break temporal
continuity.

4). The approach relies on the assumption that image and weight manifolds are related through an orthogonal transform plus a linear projection. This is a strong assumption, but the paper provides only empirical evidence. I would like to know can authors include at least some theoretical or intuitive justification for why this correspondence arises.

5). Many of the new works on INRs have not been cited. specifically, video inr methods.

6). For image-based evaluation, why PCAinit is not compared with meta-learning-based approach?

**Questions:**

Please see the weaknesses section.

---

> ### Author Response · Authors · 2025-11-20
> **Response to Reviewer 8d9P - Part I**
>
> > Q1: 1) I would like to know why authors specifically selected SIREN as the base architecture? Did authors attempt GAUSS, WIRE or any improved versions of SIREN (for instance FINER).
>
> A1: We thank the reviewer for this question. We agree that evaluating PCAInit on more advanced INR architectures such as GAUSS, WIRE, or FINER would be valuable and is an interesting direction for future work.
>
> In this paper, we chose SIREN as the base architecture mainly for two reasons. First, SIREN is a widely used, well studied, and well understood baseline for INRs, with a simple and standardized architecture and initialization scheme. This makes it easier to isolate and analyze the effect of our proposed initialization method without confounding factors from more complex designs. Second, our main focus is on understanding and exploiting the alignment between image space and weight space, rather than on proposing a new INR architecture. Using SIREN allows us to build directly on a well established setting and to make our results comparable to prior work.
>
> Furthermore, conceptually, PCAInit does not rely on SIREN-specific details and should be applicable to other INR variants that share a similar fully connected structure. We will clarify this rationale in the paper and explicitly mention extending PCAInit to GAUSS, WIRE, FINER, and related architectures as promising future work.
>
> ---
> > Q2: 2) As this paper focuses on improving initialization for video frames, how does PCAInit compare to architectures explicitly designed for videos, such as the NeRV family or related INR models? Can PCAInit also be applied to those networks, or are there structural limitations?
>
> A2: We would like to clarify that our current work focuses on per-frame SIREN-based INRs in order to cleanly study the alignment between image space and weight space and to isolate the effect of the proposed initialization. Architectures such as NeRV and related video-specific INRs are designed for a different setting, where a single network represents the entire video. A direct head-to-head comparison is therefore not straightforward and would require a dedicated experimental setup that is beyond the scope of this paper.
>
> We would like to emphasize that PCAInit is not tied to SIREN itself but to the idea of performing PCA over a collection of representations (for example frames, features, or weights) and using the resulting components to construct an initialization in weight space. In principle, a similar idea could be adapted to NeRV-like architectures, for instance by applying PCA on intermediate feature representations or on the parameters of a video INR and then unprojecting back to network parameters.
>
> However, such an extension would require careful architectural design, since NeRV-style models contain convolutional, upsampling, and positional encoding blocks rather than the simple fully connected structure of SIREN. Investigating how to best integrate PCAInit into these more complex video-specific INRs is an interesting and non-trivial research direction, and we consider it a promising direction for future work. We will clarify this discussion in the revised manuscript.
>
> ---
> > Q3: 3) The evaluation is limited to 15 videos from DAVIS 2017, which are all natural videos. It would be valuable to understand how the approach performs on non-natural or AI-generated videos, or on videos with abrupt scene transitions that break temporal continuity.
>
> A3: We thank the reviewer for this helpful comment. Regarding videos with abrupt scene transitions that break temporal continuity, this setting is closely related to our experiment in Section 4.2 ("Image-collection evaluation"), where each frame in the sequence can come from a completely different scene. In this case, there is effectively no temporal continuity between consecutive frames. The results in Table 2 show that PCAInit still achieves strong reconstruction performance under this challenging configuration, suggesting that PCAInit is robust even when temporal continuity is severely disrupted, and therefore we expect it to work well on videos with abrupt scene transitions.
>
> **Additional experiments.** Regarding non-natural or AI-generated videos, we agree with the reviewer that evaluating PCAInit on such data is important to better understand its generalization. Motivated by the reviewer's suggestion, we are currently running additional experiments on surgical videos and AI-generated video clips. Since our method does not rely on any domain-specific assumptions, it should in principle work in such scenarios. We have reported these additional experiments in the revised appendix.

---

> ### Author Response · Authors · 2025-11-20
> **Response to Reviewer 8d9P - Part II**
>
> > Q4: 4) The approach relies on the assumption that image and weight manifolds are related through an orthogonal transform plus a linear projection. This is a strong assumption, but the paper provides only empirical evidence. I would like to know can authors include at least some theoretical or intuitive justification for why this correspondence arises.
>
> A4: We thank the reviewer for raising this important point. We agree that the assumption about the relation between image and weight manifolds is strong and that our current support is mainly empirical. In this paper, our work is driven by an empirical discovery, and a full theoretical treatment is still ongoing, as mentioned in the conclusion.
>
> We can, however, provide an intuitive justification for why such a correspondence may arise. A related observation appears in the "Learning a Space of Implicit Functions" section of the SIREN paper, where a hypernetwork (\Psi) maps a latent code vector (z) to the weights of a SIREN in order to reconstruct signals from sparse pixel observations. The authors conclude that generalization over SIREN representations is at least as powerful as generalization over images. This suggests that the set of SIREN weights that represent natural images forms a structured, low-dimensional manifold that is closely related to the manifold of images itself by an approximate isometry.
>
> In our setting, each frame is well represented by a SIREN with a fixed architecture, and optimization moves the weights within a low-dimensional subset of parameter space that corresponds to the observed variability in the images. When we apply PCA to both the image space and the corresponding weight space, PCA extracts the main directions along which frames vary. Since the same underlying content changes drive both the pixel values and the optimized weights, it is natural that the principal components in the two spaces align up to an approximately orthogonal transform and a linear projection, which is what we observe empirically.
>
> We will clarify this intuition in the revised manuscript and explicitly state that our current assumption is a working hypothesis supported by empirical evidence, while a complete theoretical analysis is left for future work.
>
> ---
> > Q5: 5) Many of the new works on INRs have not been cited, specifically video INR methods.
>
> A5: The reviewer is indeed correct that we have missed several important works in the current version of the paper. In the revised version, we will expand our related work section to include recent INR methods, with a particular emphasis on video INR approaches.
>
> ---
> > Q6: 6) For image-based evaluation, why PCAInit is not compared with meta-learning-based approach?
>
> A6: **Additional experiments.** Motivated by the comment of the reviewer, in the revised version we have added a meta-learning-based baseline to Table 2 for the image-based evaluation and discuss the comparison in the text. We are pleased to report that our method outperforms the meta-learning-based baseline as well.

---

### Official Review · Reviewer_dg3F · 2025-11-04

**Soundness:** 3
**Presentation:** 3
**Contribution:** 3
**Rating:** 4
**Confidence:** 5

**Summary:**

In this work the authors introduce a new method to initialize Video-INR systems to encode the videos much faster than the previous baselines. They introduce a novel training free PCA-based initialization method which outperforms MAML based methods and is also void of any sequential dependency. This work paves way towards more practical and faster encoding times for Video INRs.

**Strengths:**

- The paper is well written and easy to follow with most claims backed by theoretical/empirical analysis.
- The idea of obtaining weight space initialization by using pseudo weight trajectories without any expensive MAML training is an important contribution to the field.

**Weaknesses:**

- Initialization strategies like re-using previous frame/first frame have been explored in prior works like [1] and the authors should refrain from claiming it as "main contributions" in their paper.
- Impact on Compression: Apart from serving as general purpose representations, INRs can also be used for video compression. It would be interesting to see the impact of PCAInit on reducing the bits required for representing a video.
- More analysis on how the content of the video used for calculating PCAinit influences the convergence speed/final quality would be helpful. That would help us answer few crucial questions like - does this transfer across datasets of videos? How does PCAInit fare when the initialization is derived from a completely different set and so on.
- The paper restricts itself to 480p videos. Does it have the potential to scale ? or does that require substantial architectural changes?
if it is the latter, then we might even need to revisit the assumptions made to derive the PCAinit weights.


[1] https://arxiv.org/abs/2212.14593

**Questions:**

NA

---

> ### Author Response · Authors · 2025-11-20
> **Response to Reviewer dg3F - Part I**
>
> > Q1: Initialization strategies such as re-using the previous frame or the first frame have been explored in prior works like~[1], and the authors should refrain from claiming this as one of the "main contributions" of their paper.
>
> A1: We thank the reviewer for pointing this out. It is true that our initialization strategy of re-using the previous frame is related to prior work. However, NIRVANA~[1] fits SIREN in a *feature space*, and the resulting features are then processed by a NeRV-style block to produce RGB pixel values, whereas our method directly fits SIREN in the *image space*. To the best of our knowledge, this direct image-space initialization and reconstruction setting has not been explored in NIRVANA.
>
> In line with the reviewer’s comment, we will revise the paper to (i) more clearly acknowledge the connection to NIRVANA, (ii) avoid overstating the novelty of the general idea of using previous-frame or first-frame initialization, and (iii) explicitly highlight that our contribution lies in demonstrating and analyzing this strategy in the direct image-space SIREN setting.
>
> ---
> > Q2: Impact on Compression: Apart from serving as general purpose representations, INRs can also be used for video compression. It would be interesting to see the impact of PCAInit on reducing the bits required for representing a video.
>
> A2: We appreciate the reviewer for raising this point. Indeed, INRs are promising for video compression, and studying the effect of PCAInit on bit rate would be an interesting research direction. However, compression is outside the scope of the current work.
>
> In this paper, our focus is on understanding and exploiting the alignment between image space and weight space in order to design better initialization strategies for INRs. We therefore evaluate PCAInit primarily through reconstruction quality (for example PSNR and SSIM) on video frames rather than a full rate–distortion analysis. A thorough compression study would require a dedicated experimental setup, including a specific entropy coding scheme for the INR parameters, a well-defined bit rate budget, and comparisons with existing video codecs and INR-based compression methods.
>
> We agree that combining PCAInit with an INR-based compression pipeline is a valuable research direction, and we will add a discussion of this in the paper and leave a detailed investigation of the compression impact of PCAInit to future work.
>
> ---
> > Q3: More analysis on how the content of the video used for calculating PCAInit influences the convergence speed/final quality would be helpful. That would help us answer few crucial questions like – does this transfer across datasets of videos?
>
> A3: In the current version of the paper, we only partially address this question by varying the content of the videos used in our experiments (see Appendix A.1). Across these diverse videos, we observe that PCAInit consistently achieves stable and high reconstruction quality, which suggests a certain robustness to video content diversity.
>
> In Figure 5 of the main paper, we report the final PSNR values for 15 different videos. These results show that the final reconstruction quality is consistent across videos with different content characteristics, as reflected by similar median values and interquartile ranges.
>
> **Additional analysis.** Motivated by the reviewer's comment, we have further analyzed the impact of video content on convergence behavior. In the revised paper, we have added an additional figure in the appendix (subsection "PCAInit per epoch") that shows the PSNR evolution over optimization epochs for all 15 videos when using PCAInit. This analysis confirms that convergence speed is also consistent across diverse types of video content.

---

> ### Author Response · Authors · 2025-11-20
> **Response to Reviewer dg3F - Part II**
>
> > Q4: How does PCAInit fare when the initialization is derived from a completely different set and so on.
>
> A4: We thank the reviewer for this interesting suggestion. We agree that it is important to understand how PCAInit behaves when the initialization is computed from a different video than the one being reconstructed.
>
> Due to computational constraints (training one model takes approximately one day in our setup), we did not explore this setting in the original submission. Motivated by the reviewer's comment, we are now running an additional experiment in which PCAInit is computed from the *bear* video and then used to initialize the model for the *bike-packing* video.
>
> Our preliminary investigation revealed that PCAInit also results in an improvement in such scenarios as well. We have reported these results in the revised appendix and discussed how cross-video initialization affects convergence and final reconstruction quality.
>
> We again thank the reviewer for pointing out this promising direction for further experiments.
>
> ---
> > Q5: The paper restricts itself to 480p videos. Does it have the potential to scale? Or does that require substantial architectural changes? If it is the latter, then we might even need to revisit the assumptions made to derive the PCAInit weights.
>
> A5: The reviewer is right to point out this restriction. In the current submission, we restrict our experiments to 480p videos mainly due to computational constraints, since training each model is already time-consuming at this resolution. Unfortunately, Full HD resolution would require substantially longer training times on our hardware setup, which is not feasible.
>
> With that being said, conceptually, PCAInit itself is not tied to a specific spatial resolution. The procedure (computing PCA on flattened frames and using the result $\mathbf{W}^\mathbf{V}$ to initialize) can be applied to higher resolutions without changing the network architecture. The main practical challenges in scaling up are computational and memory related, namely:
>
> * the cost of performing PCA on higher dimensional frame vectors, and
> * the memory required to store the weights of all frames when computing the PCA-based initialization.
>
> These limitations are due to resource constraints rather than fundamental issues with PCAInit and can be alleviated by using an appropriate method for PCA in high-dimensional spaces.

---

### Author Response · Authors · 2025-12-03
**Summary of Revisions**

We once again thank all reviewers for their helpful comments, and we are deeply sorry for not being able to have a fruitful discussion due to the recent events. Despite the circumstances, we have tried our best to answer all questions point by point in the individual responses and revised our paper accordingly. To assist the reviewing process, we have also highlighted changes in blue font in the revised version.

To further assist the Area Chair and to summarize the modifications one more time, we provide a recap of changes made in the manuscript according to the feedback we received:

* Revised the abstract to highlight our findings related to the alignment between image and weight spaces.
* Further revised the abstract to highlight our final proposed initialization (PCAInit).
* Clarified the relation between our “Previous Frame” method and NIRVANA.
* Clarified that the main focus of the paper is weight initialization, and that we use video frame reconstruction primarily as a testbed for our analysis.
* Revised the discussion of INR-based video representation methods (e.g., NeRV) in the related work section.

We have also added new experiments as well as new results in the appendix, demonstrating:

* PCAInit applied to a single image;
* Removing SIREN initialization from PCAInit;
* PCAInit applied to non-natural and AI-generated videos;
* Cross-video initialization with PCAInit.

---

### Meta-Review · Area_Chair_VVrU · 2026-01-07

**Summary:**

While the paper is clearly written and presents an interesting observation about the relationship between image and weight spaces, which leads to a simple PCA-based initialization strategy, the overall contribution appears incremental. Several reviewers note that similar initialization strategies have been explored previously, and the claimed novelty is therefore limited. Important empirical gaps also remain: the evaluation is restricted in scope (e.g., mainly 480p natural videos from a small dataset), the impact on bitrate is unclear, and the method is not compared against stronger or more modern video-specific INR architectures. In addition, some key assumptions lack theoretical justification, and aspects of the methodology are not always described with sufficient precision. Taken together, these concerns weaken the case for acceptance despite the paper’s positive qualities.

**Reviewer Concerns:**

As far as I can see, the authors have made a thorough attempt to address the reviewers’ concerns. However, it is unclear whether the reviewers will be convinced. For example, with respect to the first weakness raised by Reviewer dg3F, even after reading the authors’ response, I still feel that the proposed initialization strategy is only a minor variant of NIRVANA.

**Reviewer Scores:**

Reviewer dg3F: 4
Reviewer 8d9P: 6
Reviewer QzFL: 4
Reviewer pSrS: 8

I anticipate that the reviewers will retain their original scores. I would also note that, although Reviewer pSrS assigned a positive rating, his/her review is relatively brief compared with the others.

---

### Decision · Program_Chairs · 2026-01-26

Reject